# Novel Multibus Multivoltage Concept for DC-Microgrids in Buildings: Modeling, Design and Local Control

**Heriberto Rodriguez-Estrada** [1,*], **Elias Rodriguez-Segura** [2], **Rodolfo Orosco-Guerrero** [2], **Cecilia Gordillo-Tapia** [2] **and Juan Martínez-Nolasco** [3]

1 Doctorado en Ingeniería Electrónica, Tecnológico Nacional de México/IT de Celaya, Celaya 38010, Mexico
2 Departamento de Ingeniería Electrónica, Tecnológico Nacional de México/IT de Celaya, Celaya 38010, Mexico
3 Departamento de Ingeniería Mecatrónica, Tecnológico Nacional de México/IT de Celaya, Celaya 38010, Mexico
* Correspondence: heriberto.rodriguez@itcelaya.edu.mx

**Abstract:** In this paper, a novel microgrid (MG) concept suitable for direct current (DC) multibus architectures is depicted. Multibus feature is improved in order to distribute power in DC using a number of buses at different voltage level. A teachers offices building that houses several kinds of loads, including a charging station for electric vehicles (EV), is considered to validate the strategy. Several topologies of power electronics converters (PECs) are included in the system to perform specific tasks and providing isolation between bus and final loads. In order to develop the PECs, first, a switching function is used to obtain average model of each converter. Then, converters design is done by using well known methods that allow to obtain parameter values of all the devices in every version of each kind of converter. A hierarchical control is selected to govern the direct current microgrid (DCMG). At a lower control level, local control stage is implemented and tuned using models and designs obtained, with linear controllers in some PECs and classic strategies in others. In higher control level, there is a supervisory strategy that prioritizes the use of generated power to supply the building's loads. This energy management system (EMS) is based in Petri net theory; it consists of a start-up test, then source condition synchronous algorithm and load condition synchronous algorithm operate the DCMG according to the mentioned priority. Finally, PECs are tested on standalone, performing in closed loop, facing load changes to verify the adequate operation. Some trajectories of a simplified version of the CDMG are tested with local control in order to validate the multibus multivoltage concept. In order to verify coordinated control, some events managed by EMS are presented.

**Keywords:** direct current microgrid; multibus; multivoltage; space state model; feedback state controller; hierarchical control, electric vehicle charge station; energy management system; Petri net

## 1. Introduction

More than a decade ago, news about fossil fuel resources reduction in underground layers of the Earth appeared; at the same time, human concern about environmental problems has increased. Taking care on these issues, the electric power systems evolve in two ways. First, the vision of a classic centralized generation turns into distributed generators (DG) development and commissioning. DGs are able to supply ≤100 kW nominal power and most of them operate from renewable energy sources (RES) such as eolics, photovoltaics, hydroelectric, geothermal and biofuel. Second, nowadays, distribution aims are: reliability, high efficiency and able to meet demand response in spite of daily aging infrastructure. The main objective is to supply commercial spaces, industrial sites, offices buildings and residential places at any geographical area with no disturbances.

Besides of DGs, a modern power system includes energy storage systems (ESSs) based in energy storage units (ESUs) such as batteries or supercapacitors to support the intermittent nature of some RES such as photovoltaics or eolics. The aim of this set is to

supply power to different types of loads; some of them can be called recent in the market, such as electric vehicles (EV). The concept that considers DGs and ESSs operating together to supply power to loads appeared just over a decade ago. In order to make this idea come true, PECs are included to provides control and flexibility to a general structure called MG [1].

For any MG, it is possible to define three general objectives: higher reliability, losses reduction in feeders and improved efficiency in local supply. In order to reach them, three critical components are needed as part of the MG [2]. They are: local control in each PEC, high level control layers or/and EMS and protections array. In this manuscript, we describe the beginning stage of a long-term development; the final goal is the implementation of a MG placed in a teachers offices building for experimental purposes. Specifically in this paper, a novel concept for a MG architecture is presented with in situ linear controllers are proposed and includes a supervisory control level with a EMS based in Petri net theory [3]. Protections arrays are not part of this document.

An MG can be classified according to two criteria. First, the voltage at common connection point (CCP), power distribution could be in alternating current (AC) or in DC [4], also hybrid architectures have been presented in literature. The second criteria is about the system operation. If the MG interacts with mains, it is called interconnected mode; else, when MG is not connected to mains, it is said that the MG is operating in island mode [5]. Some systems are able to change its operation mode eventually. When the MG elements are integrated in a DC bus, parameters such as reactive power flow, frequency variations and amplitude disturbances do not take place. The result in the MG is a less complex system to control [6]; even a higher efficiency is reached in DC distribution [7]. These are the reasons to select a DCMG able to interacts with mains for this development.

PECs are an important part of the DCMG architecture because they add control and flexibility to the system to work in a harmonic way. In order to reach this system condition, the most popular option is to implement a coordinated control structure. Agreed with norm ISA-95, several layers of control can be established in a hierarchical control structure [8]. Operations such as voltage and current regulation, maximum power point tracking (MPPT), battery state of charge (SoC) estimation and some other basic functions take place at zero level [9,10] . At level number one: droop control, virtual impedance loop and power calculation that can be performed with variables measured and processed in the own converter [11–13]. The local control stage includes level zero and level one of the hierarchical control structure. In higher levels, supervisory control brings extended functions to MG such as: secondary control to eliminate voltage deviations, tertiary control to manage connection–disconnection events with mains and the ability to change the operation mode in a specific converter [14,15]. All of these tasks can be combined as algorithms of an EMS, which fulfill one of several specific system purposes. Coordinated control is reached when local control and supervisory control works together.

A popular option to implement zero level of local control is a cascade control configuration with an internal current loop and a external voltage loop. Control cascade schemes are reported using different strategies for controllers; proportional–integral (PI) classical controller is the most popular because is easy to tune, brings zero error in steady state and adds robustness [16]. Some other designs for loops are based in: classical proportional–derivative (PD) as filter [17], fuzzy logic [18] and boundary controller [19]. For level one, there are two possible techniques for droop control design: select power as signal output [5] or choose current as feedback signal [20]. In droop control strategy, droop coefficients influence system stability and current sharing, and there is a trade-off between current sharing and voltage bus deviation [21,22]. Sometimes, a virtual impedance loop, droop controller and power calculator strategy can be included. In this paper, a linear technique is selected to design a state feedback as local controller to reach well regulated buses, to support equally shared current distribution and get over input voltage variations; all of these issues are possible because all the dynamics in the PEC feedbacked.

Once the local control stage has been defined and taking into account the benefits of state feedback control, an EMS is included as a higher coordinated control level. Coordinated control provides an improved power collected manage, and bus stability can be enhanced [23]. An EMS allows to harness power collected from DGs to the loads in a defined way. The DCMG structure and the EMS design can be diverse, and they are made up in accordance with the specific objectives that are desired to be achieved. Some of EMS are reported perfoming basic tasks such as management of the energy between source and load based on optimal scheduling [24] and power flow management depending on ESS batteries Soc [25], using virtual impedance [26], or based in fuzzy logic [27]. In this paper, a novel EMS is focused to harness all the power harvested in PV arrays to supply the loads, to charge the ESS batteries or to inject to mains is presented. The EMS is conceived in a similar way to an industrial production system, under the premise that all of the energy that is produced in the DGs is consumed in its entirety. The EMS is modeled as a Petri net and has three stages: startup test, source condition establishment stage and load condition establishment stage.

In the MG proposed architecture for teachers building, an electric vehicle charge station (EVCS) is included. The EVCSs can be classified in three groups according to the input voltage and the power rating [28–30]. Table 1 shows the information ordered and summarized. Then, a commercial sigle phase AC-EVCS is an adequate selection to be supplied by the highest voltege level bus of the proposed DCMG.

**Table 1.** Classification of charge stations based on power levels.

| EVCS Type | Power Supply | Charger Power | Charging Time |
|---|---|---|---|
| AC L1 residential | 120/220 $V_{AC}$, 12–16 A, single phase | 1.44–1.92 kW | Approx. 17 h |
| AC L2 commercial | 208–240 $V_{AC}$, 15–80 A, single/split phase | 3.1–19.2 kW | Approx. 8 h |
| DC L3 fast charger | 300–600 $V_{DC}$, *Max* 400 A, | 120–240 kW | Approx. 30 min |

The objective of manuscript is to validate the design and performance of a novel concept for DCMG called multibus multivoltage architecture; it is the starting stage of an design suitable to supply power, voltage and current requirements of loads inside a teacher's offices building. In Section 2, the DCMG particular architecture is presented in a detailed way. In Section 3, the procedure to obtain an average model for all of the PECs in the system using a particular switching function is depicted. In Section 4, design procedures for PECs are mentioned. The aim is to size the passive elements of each converter version using typical design methods; inductor current ripple into a fixed limit criteria is set to avoid high frequency components propagation when a PECs cascade connection occurs. In Section 5, the development of the local control stage for every converter is shown; the innovation of using a linear control technique based on feedback state controllers that allows to keep the output voltage regulated, good response to transients and favors equal current distribution from source converters to loads, then a three stage EMS that harness all the power collected in PV to supply loads in the place is included. Section 6 includes simulation results for standalone converter with load changes; also, some critical power trajectories for the proposed DCMG are enabled to test local control, and some events managed by EMS are presented to validate the concept. Discussion and conclusions are presented in Section 7.

## 2. Proposed MG Architecture

About the DCMG scheme, multibus architectures, according to the original concept, are able to perform with higher reliability, basically because there is more than one bus in the system to overpass a fail [31]. Each bus gathers at least one DG, one ESS and some loads.

The proposed system must be able to satisfy the power demand in the teachers offices building as well as to supply an EVCS or, eventually, it can be able to inject power to mains. Some months ago, a quality energy logger equipment was installed to quantify power consumption. At the same time, a loads characterization study was performed to identify voltage and current requirements at every load inside the building. The highlighted results are: the bulding peak power consuption is 6 kW; in the other hand, it is necessary to set three DC buses at different voltage level because of the different type of the loads in the building. A primary bus at 380 V to supply electronic loads at every office; another secondary bus at 190 V to power the lighting system of the place (this bus can support the primary bus as an emergency action eventually). From the secondary bus, unidirectional 48 V buses are derivated; every one of them supplies six tandems driver-LED luminary for a specific area lighting. The primary bus and the secondary bus each incorporate an ESS based in batteries. At this point, it can be noticed that the MG has a particular architecture; then, it can be mentioned, according to the aforementioned two criteria, as multibus multivoltage DCMG.

The selected DCMG is a flexible architecture able to satisfy the building power demand. In the place, four photovoltaic (PV) arrays are installed as DGs, each at 2.5 kW power rating, eventually 10 kW total power. Then, three buses distribute the power to every load in the building mainly using primary bus (380 V) and secondary bus (190 V). Harvested power at PV arrays is processed through boost converters—three of them linked to the primary bus and one more setting the secondary bus. This kind of structure is able to transfer power from one bus to another at a power level lower than its rated power. In typical multibus architectures, power is transfered between buses with similar level voltage using static switches, but in the DCMG proposed, primary and secondary buses are at different voltage levels. This feature is the main difference with DC typical multibus systems in [32,33]. This development uses bidirectional power converters to transfer between buses, instead of static switches; then, it is possible to set the secondary bus from the primary bus and vice versa. A couple of PECs with bidirectional feature allows batteries charge/discharge at the ESSs. From secondary bus, 48 V unidiretional buses are derived to supply six tandems driver-LED luminary, four 48 V unidirectional buses are able to supply twenty-four luminaries inside the place. In order to improved DCMG security, all of the eventual power consumers in the system use isolated converters as interface to satisfy current and voltage required by final load. The architecture can be defined as multibus multivoltage; this concept brings flexibility to design a DCMG at any place. In order to validate the concept, a simplified DCMG with all the PECs but not all the loads is shown in Figure 1.

Inside the building, there is a distribution center where most of the PECs are placed. Those are the boost converters, bidirectional PECs to transfer power between buses, bidirectional PECs to charge–discharge ESSs batteries and unidirectional converters to supply lighting system. The remaining PECs are placed at different distances from load center—in this case, line resistive–inductive impedance effects must be taken into account. Line capacitance is not significant and can be neglected. Inductance and resistance values for THW-LS wire are calculated according to expressions in [34]; results for different lengths appear in Table 2.

The solar panel used for DGs, as PV arrays, is a similar version to part number JC250M-24/Bb, series Virtus II, manufactured by ReneSola. PV panel simplified model parameters are shown in Table 3.

This architecture concept allows to add some other kind of DGs. Then, addition of new DGs can increase the system complexity, but the performance and controllers can mainly be kept.

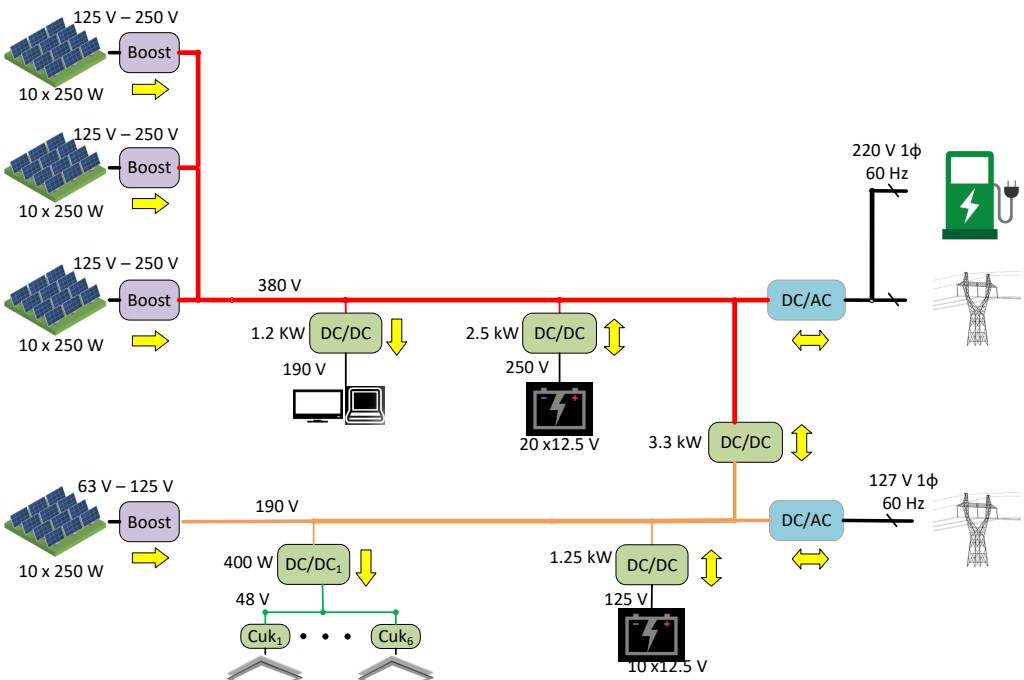

**Figure 1.** Multibus multivoltage DCMG architecture proposed.

**Table 2.** Line impedance values for different wire gauges and lenghts.

| Lengh (Meter) | AWG 14 (μH; mΩ) | AWG 12 (μH; mΩ) | AWG 10 (μH; mΩ) | AWG 8 (μH; mΩ) |
|---|---|---|---|---|
| 5 | 4.53; 1.71 | 4.29; 1.62 | 4.06; 1.53 | 3.83; 1.44 |
| 10 | 9.06; 3.41 | 8.58; 3.23 | 8.12; 3.06 | 7.66; 2.89 |
| 15 | 13.59; 5.12 | 12.87; 4.85 | 12.18; 4.59 | 11.49; 4.33 |
| 20 | 18.12; 6.83 | 17.16; 6.47 | 16.25; 6.12 | 15.32; 5.77 |
| 25 | 22.65; 8.53 | 21.45; 8.08 | 20.31; 7.65 | 19.15; 7.22 |

**Table 3.** Solar panel model parameters.

| Parameter | Value |
|---|---|
| Maximum power $P_{max}$ | 250 W |
| Current at maximum power $I_{mp}$ | 8.31 A |
| Voltage at maximum power $V_{mp}$ | 30.1 V |
| Short circuit current $I_{SC}$ | 8.83 A |
| Open circuit voltage $V_{OC}$ | 37.4 V |

## 3. Power Converters Modelling

As already mentioned, DCMG structure contains different PEC topologies that fulfill particular functions in the system. In this section, we presents modeling developments for converters that are part of the DCMG: boost converter, DC–DC full bridge converter, Cuk converter and DC-AC $1\phi$ full bridge models are obtained. Each converter procedure of modeling appears in Sections 3.1–3.4.

### 3.1. Boost Converter Average Model

Each PV array has an interface boost converter before the connection with the corresponding bus. In this DCMG, boost converter preserves the traditional topology, but an input capacitor has been included to avoid PV array output voltage variations and a thermistor previous to the mentioned capacitor to limit the inrush current at the initial charge (see Figure 2).

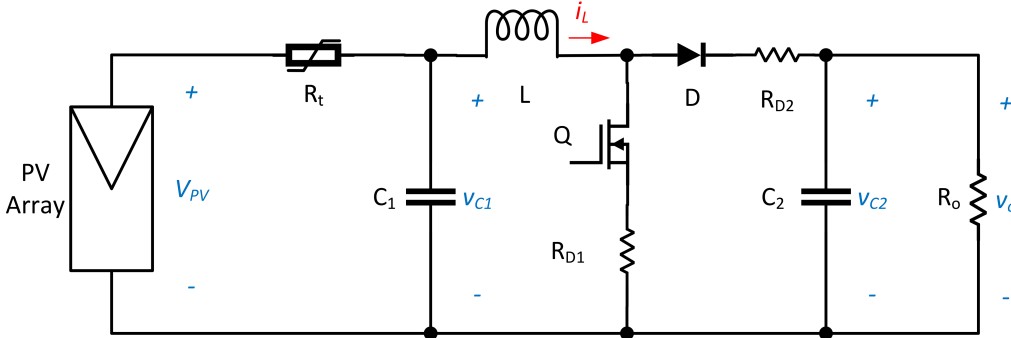

**Figure 2.** Boost converter.

In Figure 2, boost converter scheme has two power devices, a MOSFET ($Q$) and a diode ($D$); then, a switching function is defined for each in order to obtain the average model for the converter. In this way, $S_1$ and $S_2$ are switching functions for $Q$ and $D$, respectively. If $S_1 = 1$, $Q$ is in conduction state, else, if $S_1 = 0$, $Q$ is not conducting. Same logic is used for $S_2$ and $D$, but $S_1$ and $S_2$ perform in complementary mode, so the two power devices operate in a complementary way. The following combinations are not allowed $S_1 = 1$ and $S_2 = 1$; $S_1 = 0$ and $S_2 = 0$. In order to have an improved resultant model of this converter, the conduction resistances for the switching devices have been included: $R_{D1}$ for the MOSFET and $R_{D2}$ for the diode. The Kirchhoff Voltage Law (KVL) is applied to inductor loop and Kirchhoff Current Law (KCL) is used in capacitors nodes in order to obtain the state equations for the switching model. The average of the switching function, $\overline{S}(t)$, corresponds to modulator signal duty cycle, $u(t)$, in this case, $\overline{S}(t) = u(t)$. Taking this deduction into account, it is possible to go from the switching model to the averaged model, which is given by:

$$\frac{dv_{C1}}{dt} = -\frac{1}{R_t C_1} v_{C1} - \frac{1}{C_1} i_L + \frac{1}{R_t C_1} V_{PV}$$
$$\frac{di_L}{dt} = \frac{1}{L} v_{C1} - \frac{[(R_{D1} - R_{D2})u + R_{D2}]}{L} i_L - \frac{(1-u)}{L} v_{C2}$$
$$\frac{dv_{C2}}{dt} = \frac{(1-u)}{C_2} i_L - \frac{1}{R_O C_2} v_{C2} \tag{1}$$

As mentioned before, the architecture of the DCMG has two main buses in different voltage level, so the model obtained can be used for both versions, the three 380 V boost converters linked to the primary bus and the 190 V boost converter in the secondary bus.

*3.2. DC–DC Full Bridge Converter Average Model*

DC–DC full bridge converters process power from one of the main buses to meet voltage and current requirements for final loads like LED luminaries, personal computers, etc. This topology is an important part of the system because there are several versions at different power ratings in the DCMG. This configuration allows an adequate output voltage regulation, provides isolation between the bus and the final load and gives reliability to the general system. This kind of isolated converter is able to manage a greater power rating than other isolated topologies. DC–DC full bridge converter is a two-stage topology, integrated by an inverter ($Q_1$–$Q_4$), a rectifier ($D_1$–$D_4$), linked by high-frequency transformer, and a *LC* output filter (Figure 3).

The DC–DC full bridge converter operates with phase shift pulse wide modulation. Two specific pair of power devices in the inverter are defined; $Q_1$ and $Q_2$ are the first pair and $Q_3$ and $Q_4$ are the second pair. The conduction period of time for each pair of MOSFETs must last less than a half of switching period. Diodes $D_1$, $D_2$, $D_3$ y $D_4$ conduct depending on its own polarization state in a specific time. A switching function $S_1$ is defined according to the on and off times in the devices $Q_1$, $Q_2$, $D_1$ and $D_2$ take the zero state and the one state simultaneously. In the same way, $Q_3$, $Q_4$, $D_3$ and $D_4$ synchronize

their on and off state, and the corresponding switching function $S_2$ is assigned to this set; $S_1$ and $S_2$ operate with $180°$ phase. In order to obtain the switched model, $v_x$ is defined as the rectifier voltage output. By applying Kirchhoff's laws, equations for switching model in time domain are obtained from the DC–DC full bridge switching equivalent circuit. Then, the average model is obtained by considering the modulating signal, $u(t)$, as the average of the selected commutation function; the resultant average model is described as:

$$\frac{dv_{C2}}{dt} = -\frac{1}{R_o C_2} v_{C2} + \frac{1}{C_2} i_L$$
$$\frac{di_L}{dt} = -\frac{1}{L} v_{C2} + \frac{1}{L} 2nu V_i \tag{2}$$

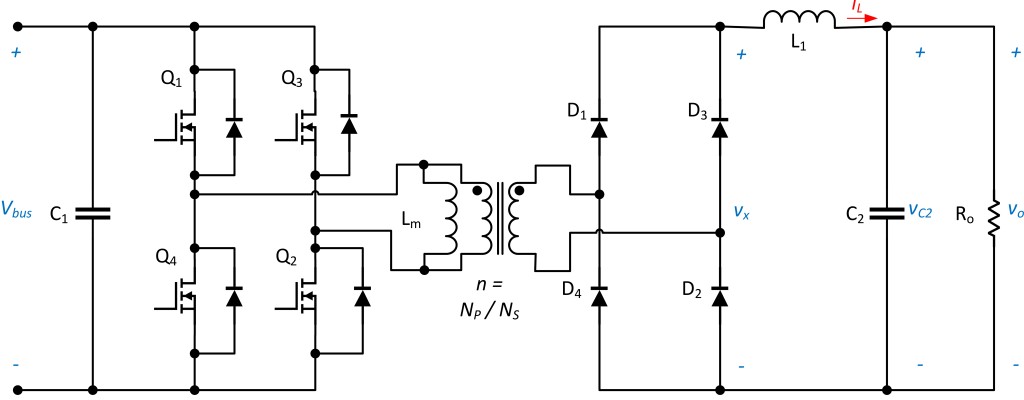

**Figure 3.** DC-DC Full Bridge Converter.

The average model in (2) can be used in any of the following versions: 380 V to 190 V at 1.2 kW to supply electronic loads in each cubicle, 190 V to 48 V at 400 W for supply drivers for LED luminaries. Two versions, 380 V to 190 V at 3.3 kW for power transfer between primary and secondary bus, in the other way, a 190 V to 380 V at 3.3 kW version. For charge–discharge bateries, a 380 V to 250 V at 2.5 kW and 250 V to 380 V at 2.5 kW for ESS at primary bus; finally, two more versions 190 V to 125 V at 1.25 kW and 125 V to 190 V at 1.25 kW for ESS in secondary bus.

*3.3. Cuk Converter Average Model*

In order to include LED luminaries as final load, a driver based on Cuk converter is designed. Cuk driver is the interface between 400 W DC-DC full bridge converter and LED luminaire; Cuk converter schematic is shown in Figure 4.

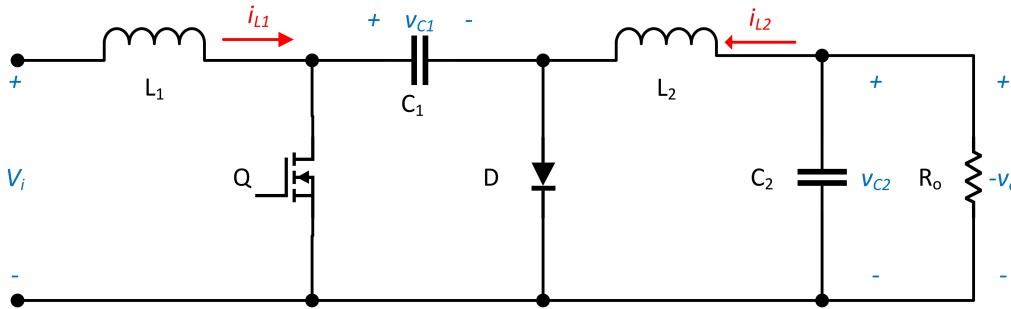

**Figure 4.** Cuk converter.

In Figure 4, two power devices, MOSFET ($Q$) and diode ($D$), can be distinguished. Switching functions $S_1$ and $S_2$ are defined for $Q$ and $D$, respectively. Again, if $S_1 = 1$, $Q$ is conducting, else, if $S_1 = 0$, the device is off state. The same logic is used for function $S_2$, and complementary operation is assumed. The following combinations are not allowed

$S_1 = 1$ with $S_2 = 1$ and $S_1 = 0$ with $S_2 = 0$. By applying Kirchhoff laws, state equations in time domain are obtained from the Cuk converter switching equivalent circuit; then, the averaged model is obtained by considering the modulating signal $u(t)$ as the average of the commutation function. The averaged model is described as:

$$\frac{di_{L1}}{dt} = -\frac{(1-u)}{L_1}v_{C1} + \frac{V_i}{L_1}$$
$$\frac{di_{L2}}{dt} = \frac{u}{L_2}v_{C1} + \frac{1}{L_2}v_{C2}$$
$$\frac{dv_{C1}}{dt} = \frac{1-u}{C_1}i_{L1} + \frac{u}{C_1}i_{L2}$$
$$\frac{dv_{C2}}{dt} = -\frac{1}{C_2}i_{L2} + \frac{1}{R_oC_2}v_{C2} \tag{3}$$

Cuk output current flowing through LED array inside the luminary must be regulated to avoid unwanted lighting effects.

### 3.4. Single Phase DC-AC Full Bridge Bidirectional Converter Average Model

Interconnected systems includes one or more PECs that interacts with mains; the topology selected has bidirectional capability. In the first operation mode, the converter takes power from the mains and supplies loads in DC side, rectifier mode. In second operation mode, power harvested in PV arrays exceeds building power demand; then, it is possible to inject power to the grid, inverter mode. Then, the selected topology is a DC–AC full bridge converter (also called H4) [35,36] and has the capability to operate bidirectionally. This feature allows the converter to be integrated in the DCMG architecture to interacts with the mains. The converter schematic diagram appears in Figure 5.

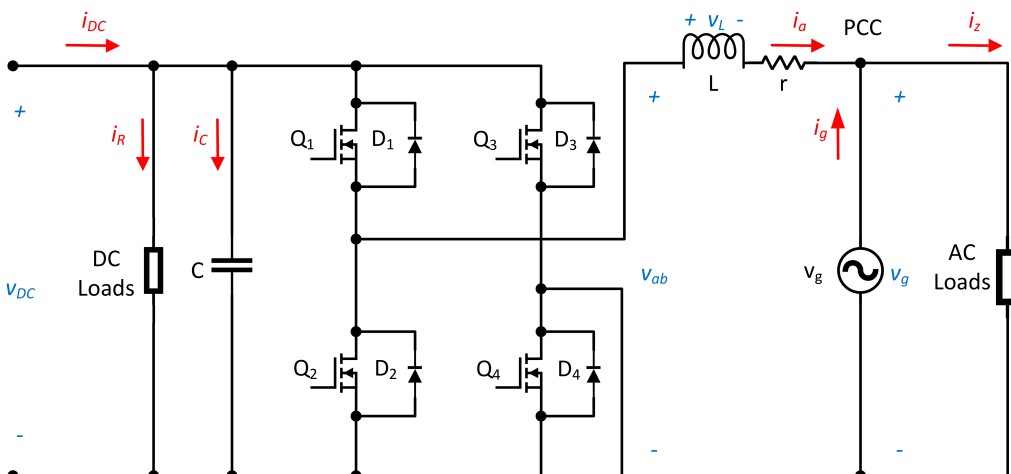

**Figure 5.** Single phase DC-AC full bridge converter.

A first pair of power devices defines is $Q_1$ and $Q_4$ and operates synchronized; the other pair $Q_2$ and $Q_3$ is synchronized as well but in a complementary way in reference with the first pair of power devices. Switching functions are defined as follows: if $S_1 = 1$, $Q_1$ and $Q_4$ are in the on state, else, $S_1 = 0$, $Q_1$ and $Q_4$ are in the off state. A second switching function, $S_2$, operates with $Q_2$ and $Q_3$ power devices. To obtain the corresponding switching simplified equivalent circuit, KVL is applied to the inductor side, and $r$ is the series resistive value associated to the inductor and contributes with some damping to system. Then, KCL is applied to the PV side; the resultant set of equations represents the converter

switching model. Finally, the average value of the switching function $\overline{S}(t)$ is defined as $u(t)$, the modulating signal.

$$\frac{di_a}{dt} = \frac{1}{L}(uv_{dc} - i_a r - v_g)$$
$$\frac{dv_{dc}}{dt} = \frac{1}{C}(ui_a - \frac{1}{R}v_{DC} + i_{DC}) \tag{4}$$

Two version of this converter are included in DCMG, 220 $V_{RMS}$ to 190 $V_{CD}$ at 3.3 kW and 127 $V_{RMS}$ to 190 V at 1.5 kW. They are placed in primary and secondary bus, respectively. The 3.3 kW version in the primary bus is able to supply the ECVE.

## 4. Converters Design

First, some considerations to take into account for converters design are: for this development system, global efficiency is not a main issue; converters design and implementation includes an output bleeding resistor at 10% of nominal power in order to avoid no-load operation and keep the converter on performing in continuous conduction mode (CCM). DCMG is an interconnection of power converters do specific tasks; eventually, some of these converters can perform in a cascade connection, and high-frequency switching noise increases at every stage until reach a unsafe condition for the system. For this reason, a design criterion is established: current ripple must be less than 10% inductor nominal current value.

### 4.1. Boost Converter Design

Using the above criteria and following the design procedure in [37], the mathematical expressions to assign values to PEC passive elements are obtained. For 2.5 kW and 380 V voltage output boost converter, eventually, up to three of these converters set primary bus and supplies EVCE. The fourth boost converter at 2.5 kW and 190 V, voltage output sets secondary bus. Passive devices minimum obtained values for $L_{min}$ and $C_{min}$, but about twenty times $L_{min}$ value is selected in order to meet ripple current criteria and keep converter operating in CCM; parameter values are shown in Table 4.

**Table 4.** Parameter design values for both versions of boost converter.

| Version ($V_{out}$) | $I_L$ (A) | $R_o$ (Ω) | $L$ (mH) | $C_1$ (μF) | $C_2$ (μF) | $R_{D1}$ (mΩ) | $R_{D2}$ (mΩ) | $\Delta i_L$ (%) | $\Delta v_{C2}$ (%) |
|---|---|---|---|---|---|---|---|---|---|
| 380 | 14.25 | 57.76 | 1.3 | 82 | 150 | 65 | 45 | 9.03 | 0.12 |
| 190 | 31.78 | 14.44 | 0.65 | 82 | 150 | 65 | 45 | 7.75 | 0.52 |

### 4.2. DC–DC Full Bridge Converter Design

For all versions of this topology, the inductor current ripple criteria is applied to avoid high frequency components in the system. In the DCMC, the different DC–DC full bridge converter versions perform particular tasks. 400 W, 190 V input and 48 V output version supplies up to six tandems driver-LED luminary from the secondary bus. 1.2 kW , 380 V input and 190 V output version takes power from primary bus to supply electronic loads, like personal computer and others in teachers offices. To carry out the transfer between the primary bus and the secondary bus, there are two more versions of the DC–DC full bridge topology, both with a capacity of 3.3 kW. The first one performs a step-down function by transferring power from the primary bus at 380 V to the secondary bus at 190 V. The second of these versions has step-up capacity and performs the opposite operation, transfering energy from the secondary bus with a voltage of 190 V to the bus primary bus at 380 V.

On the other hand, the DCMR architecture is a multibus array, so each bus has at least one GD, one ESS and some loads. In the selected system, there are two main buses, so there is one ESSs in each bus. The ESS of the primary bus is made up of a serial arrangement of twenty batteries. To carry out the power management in the ESS batteries in the primary

bus, two versions of the full bridge DC–DC converter are used, the first of these versions with a capacity of 2.5 kW is used to charge the SAE from the 380 V bus at 250 V output. The second version for this task, also in 2.5 kW to take energy from the ESS at 250 V and set the primary bus to 380 V. The ESS in the secondary bus is composed by a series arrangement of ten batteries. To carry out the power management in the ESS batteries, another couple of versions of the DC–DC full bridge converter are used; the first of these versions has a capacity of 1.25 kW and is used for the ESS load from the 190 V bus and output at 125 V. In addition, a 1.25 kW version is included in the system to take power from the ESS at 125 V and sets the secondary bus at 190 V. The design procedure is based in procedure given in [38]. All of the versions perform in 50 kHz switching frequency. The values calculated for all the versions mentioned appear in Table 5.

**Table 5.** Parameter design values for all the versions of DC–DC full bridge converter.

| Version (kW) | $V_i$ (V) | $V_o$ (V) | $I_L$ (A) | $R_o$ ($\Omega$) | $L$ (mH) | $C_1$ ($\mu$F) | $C_2$ ($C_2$) | %$\Delta i_L$ (%) | %$\Delta v_{C2}$ (%) |
|---|---|---|---|---|---|---|---|---|---|
| 0.40 | 190 | 48 | 8.33 | 5.76 | 380 | 1 | 82 | 9.03 | 0.12 |
| 1.2 | 380 | 190 | 31.78 | 14.44 | 650 | 1 | 150 | 7.75 | 0.52 |
| 3.3 | 380 | 190 | 17.37 | 10.94 | 550 | 1 | 100 | 6.09 | 0.0076 |
| 3.3 | 190 | 380 | 43.76 | 8.68 | 2.3 | 1 | 100 | 5.97 | 0.0016 |
| 2.5 | 380 | 250 | 10 | 25 | 1.3 | 1 | 100 | 6.09 | 0.0032 |
| 2.5 | 250 | 380 | 6.58 | 57.76 | 3.1 | 1 | 100 | 4.46 | 0.0011 |
| 1.25 | 190 | 125 | 10 | 12.5 | 700 | 1 | 100 | 5.43 | 0.0065 |
| 1.25 | 125 | 190 | 12.5 | 28.88 | 1.5 | 1 | 100 | 5.13 | 0.0022 |

For each version, passive elements minimum obtained values for this topology are $L_{min}$ and $C_{min}$; then, three times $L_{min}$ value to meet current inductor ripple criteria and to keep the converter performing in CCM operation.

### 4.3. Cuk Converter Design

Cuk driver is connected to one of the unidirectional tertiary buses at 48 V derived from a DC–DC full bridge converter connected to the secondary bus. The Cuk converter particular objective is to supply a LED luminary with specific voltage and regulated current level. The design is based in [38]; the resultant values appear in Table 6.

**Table 6.** Parameter values design Cuk converter.

| $P$ (W) | $f_s$ (kHz) | $L_1$ (mH) | $L_2$ (mH) | $V_{in}$ (V) | $V_o$ ($\Omega$) | $C_1$ (nF) | $C_2$ ($\mu$F) |
|---|---|---|---|---|---|---|---|
| 60 | 100 | 1.2 | 1.1 | −68 | 43.76 | 220 | 1 |

### 4.4. DC–AC Bidirectional Full Bridge Converter Design

This topology allows the interaction MG between secondary bus and mains, when solar irradiation increases until there is a power amount greater than building demand; then, the DC–AC converter is able to inject power into the grid through a single phase. Otherwise, if the PV array in not collecting power enough to fulfill loads demand inside the place, the converter sets the bus by taking power from the grid. The power stage design is based in [39]. There are two versions in the proposed DCMG. The first one is 1.2 kW to manage power transfer between secondary bus and the mains. The second version is a 3.3 kW converter. This converter transfers power between the primary bus and the mains and supplies the EVCE. The obtanied parameter values appear in Table 7.

**Table 7.** Parameter values design for single phase DC–AC full bridge converter.

| Version (kW) | $V_{ac}$ ($V_{RMS}$) | $V_{DC}$ (V) | $R_o$ ($\Omega$) | $L$ (mH) | $r$ ($\Omega$) | $C$ ($\mu$F) |
|---|---|---|---|---|---|---|
| 3.3 | 220 | 380 | 43.76 | 7 | 0.27 | 7200 |
| 1.4 | 127 | 190 | 43.76 | 4.1 | 0.2 | 4576 |

Coupling inductor is calculated for maximum active and reactive power; DC capacitor performs two functions: to filter voltage ripple in DC bus and storage power to compensate lack of active power.

## 5. Hierarchical Control Development

As mentioned before, a hierarchical control structure is selected to manage the system. In a summarized way, two main parts are distinguished: local control and coordinated control. This kind of control strategy, keeps voltage buses regulated by feedback the complete PEC state. Local control stage is able to receive enable signals, operation mode selection and references. In this manuscript, the coordinated control stage is based in an EMS that prioritizes the consumption of the power collected in PV arrays.

### 5.1. Local Controllers Development

Local control is the lowest level control stage placed in the converter that governs the PEC with in situ variable measurements. Local control is able to receive signals like references, enable and operation mode selection from higher levels controllers in order to perform coordinated control. Obtained PEC models are based in state equations in the time domain; then, the linear control design strategies are suitable for boost converter and DC–DC full bridge converters. State feedback controller uses the complete state information given by average model to obtain gain values to use, and to generate a control signal defined as $u(t)$. Outstanding features of this control strategy are: state feedback controller brings zero error in state stable and robustness to transients and input voltage variations but the design and tuning procedure is not trivial.

Cuk converter and single phase DC–AC full bridge converter perform with controllers based in classical strategies; this kind of controller is robust and easy to tune. For Cuk converter, PI controller performs current regulation, and for DC–AC converter, a cascade structure based in PI controllers with a current inner loop and voltage external loop is used for regulation.

#### 5.1.1. Boost Converters Controllers Development

Every boost converters in the system is able to perform in two modes: output voltage regulation and maximum power point tracking (MPPT). Voltage regulation strategy is used when a specific bus supplies a medium or heavy load condition. An MPPT strategy is used when there is a lot of power harvested in PV arrays and medium or light load condition is supplied by an specific bus.

*A. Voltage Regulation Controller.*

A control linear technique called state feedback is selected for local controllers design. Evaluating model boost converter obtained in (1) as a static model, i.e., with all the variables in steady state, the conclusion is the boost converter is a nonlinear system. It is necessary to obtain a linearized model in a fixed operation point. First step to linearization is to obtain matrix **A** and **B** through Jacobian for the average model evaluated in a fixed operation point. For matrix **A**:

$$\mathbf{A} = \begin{vmatrix} \dfrac{\partial f_1}{\partial v_{C1}} & \dfrac{\partial f_1}{\partial i_L} & \dfrac{\partial f_1}{\partial v_{C2}} \\[2mm] \dfrac{\partial f_2}{\partial v_{C1}} & \dfrac{\partial f_2}{\partial i_L} & \dfrac{\partial f_2}{\partial v_{C2}} \\[2mm] \dfrac{\partial f_3}{\partial v_{C1}} & \dfrac{\partial f_3}{\partial i_L} & \dfrac{\partial f_3}{\partial v_{C2}} \end{vmatrix} \begin{matrix} u = D \\ v_{C1} = V_{C1} \\ i_L = I_L \\ v_{C2} = V_{C2} \end{matrix} \tag{5}$$

For matrix **B**:

$$\mathbf{B} = \begin{vmatrix} \dfrac{\partial f_1}{\partial u} \\[2mm] \dfrac{\partial f_2}{\partial u} \\[2mm] \dfrac{\partial f_3}{\partial u} \end{vmatrix} \begin{matrix} u = D \\ v_{C1} = V_{C1} \\ i_L = I_L \\ v_{C2} = V_{C2} \end{matrix} \tag{6}$$

Particular structure of matrix **A** in linearized model obtained is:

$$\mathbf{A} = \begin{bmatrix} \dfrac{-1}{R_t C_1} & \dfrac{-1}{C_1} & 0 \\[3mm] \dfrac{1}{L} & \dfrac{[(R_{D1} - R_{D2})D + R_{D2}]}{L} & \dfrac{-(1 - D)}{L} \\[3mm] 0 & \dfrac{(1 - D)}{C_2} & \dfrac{-1}{C_2 R_O} \end{bmatrix} \tag{7}$$

and for matrix **B**

$$\mathbf{B} = \begin{bmatrix} 0 \\[2mm] -\dfrac{[(R_{D1} - R_{D2})D + R_{D2}]i_L}{L} + \dfrac{V_{C2}}{L} \\[3mm] \dfrac{-I_L}{C_2} \end{bmatrix} \tag{8}$$

Now, matrix **A** and **B** are incorporated to linealized state equation:

$$\widehat{\dot{x}} = \mathbf{A}\widehat{x} + \mathbf{B}\widehat{u}$$

with:

$$\widehat{\dot{x}} = \begin{bmatrix} \dfrac{\partial \widehat{v}_{C1}}{\partial t} \\[3mm] \dfrac{\partial \widehat{i}_L}{\partial t} \\[3mm] \dfrac{\partial \widehat{v}_{C2}}{\partial t} \end{bmatrix}$$

and:

$$\widehat{x} = \begin{bmatrix} \widehat{v}_{C1} \\[2mm] \widehat{i}_L \\[2mm] \widehat{v}_{C2} \end{bmatrix}$$

Input vector $\widehat{u}$, in this case, has a single element $u$, it corresponds to modulating signal. The output equation is given by:

$$\widehat{y} = \mathbf{C}\widehat{x} + \mathbf{D}\widehat{u}$$

For this application, direct transfer matrix has not elements $\mathbf{D} = 0$, and matrix $\mathbf{C}$ is defined as:

$$\mathbf{C} = \begin{bmatrix} 0 & 0 & 1 \end{bmatrix}$$

State feedback controller structure for boost converter is shown in Figure 6.

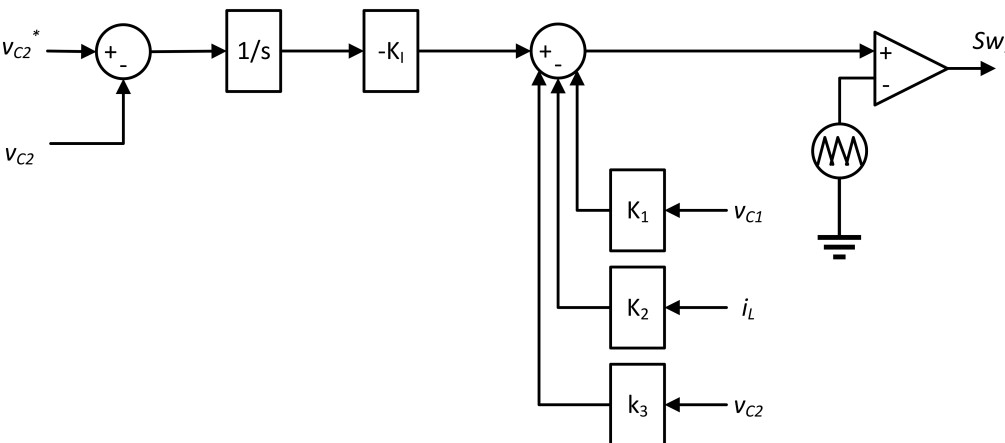

**Figure 6.** Feedback state controller structure.

Control scheme includes a gain-integrator trajectory to lead steady state error to zero and gain trajectories to feedback system state. The first step is to build up the augmented system, and the controllability matrix is required. For the second step, it is necessary to calculate gains values; the Ackermann method is applied to define gain values $K_I$, $K_1$, $K_2$ and $K_3$ as elements of the vector:

$$\mathbf{K_a} = \begin{bmatrix} K_1 & K_2 & K_3 & K_I \end{bmatrix}$$

Parameters such as overshoot percentage, %*OS*, and settle time, $T_s$, are considered to calculate the desired polinomy. Two auxiliar poles are necessary to balance system order and the gain vector $\mathbf{K_a}$. The product of desired polynomial evaluated in terms of the augmented system and controllability matrix brings out spacific values for gains in matrix $\mathbf{K_a}$. Parameter values for desired polynomial of boost converter version 2.5 kW at 380 V output are: %*OS* = 0.1% and $T_s$ = 12 ms, two auxiliar poles are needed to balance the augmented system, the first one is placed one hundred times real part of the complex conjugate value of the dominant poles, the second pole is placed two hundred times real part of the complex conjugate value. The procedure is done again in order to obtain 190 V boost converter gain values. Selected parameters to calculate desired polynomial are: %*OS* = 0.1% and $T_s$ = 8 ms. Two auxiliar poles are needed to balance. The first one is placed seventy-five times real part of the complex conjugate value of the dominant poles. The second pole is placed one hundred and fifty times real part of the complex conjugate value of the dominant poles of the complex conjugate value.

*B. MPPT Controller.*

In specific periods of time during the day, particularly in sunny seasons of year, sun irradiance can increase; then, PV arrays harvest a surplus power. Under these conditions, it is possible to perform a better harness of this increased power collected by using a MPPT algorithm. There are many strategies to implement a MPPT algorithm; the most accepted of them are summarized in [40]. Nevertheless, the most popular algorithm and the one selected for this development is called perturb and observe (*P&O*); this is a sampling technique that uses two variables measurement, current and voltage, to calculate power.

*P&O* can be implemented in analog version or in digital version and does not require a tuning procedure.

Voltage and current PV array output measurements are used to calculate a currente power data is calculated $P_1$; then, a voltage little variation ($\Delta V$) or a duty cycle disturbance ($\Delta D$) in boost converter leads to a new power value calculation, $P_2$. A comparision between $P_1$ y $P_2$ is performed. If $P_2$ is greater than $P_1$, disturbance moved in the right direction; else, disturbances must move in the opposite direction. In this way, it is possible to obtain the power peak value ($P_{mpp}$) and, consequently, voltage peak value ($V_{mpp}$). Algorithm flowchart of the MPPt controller is shown in Figure 7.

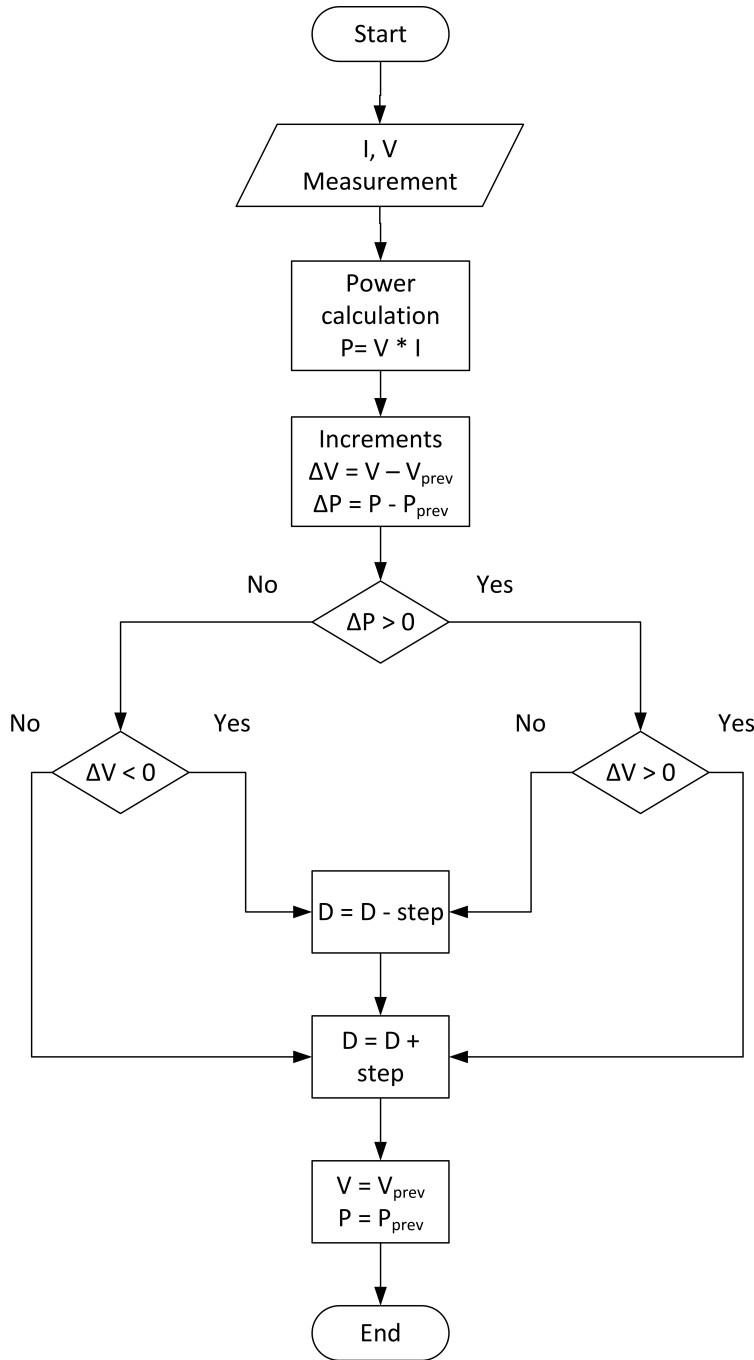

**Figure 7.** MPPT controller flowchart algorithm.

Behavior of boost converters, the two versions, performing in both modes: voltage regulation mode, in closed loop, facing load changes and MPPT controller perfomance are presented and commented in next section.

### 5.1.2. DC–DC Full Bridge Converters Controllers Development

The DC–DC full bridge converter model obtained in (2) is evaluated as a static model, i.e., with all the variables in steady state, and the obtained result is this kind of PEC is a linear system; then, a linearization procedure is not necessary. Once again, same control linear technique is used to design a feedback state controller, its structure is similar that shown in Figure 6, but, in this case, only two variables are feedbacked ($v_{C2}$ and $i_L$). Controller structure includes a gain-integrator trajectory that leads steady state error to zero and other gain trajectories to feedback system state. Furthermore, it is necessary to calculate gain values $K_I$, $K_1$ and $K_2$; they are defined using the Ackermann method. Gains before mentioned are part of the gain vector given as:

$$\mathbf{K_a} = \begin{bmatrix} K_1 & K_2 & K_I \end{bmatrix}$$

The main parameters to design desired polynomial are overshoot percentage %*OS* and settle time $T_s$. Now, one auxiliar pole in needed to balance system order and gains vector $\mathbf{K_a}$. The product of desired polynomial evaluated in augmented system matrix and controllability matrix results values for gain matrix $\mathbf{K_a}$. The selected values for desired polynomial for DC–DC converter 400 W at 48 V are: %*OS* = 0.1% and $T_s$ = 700 µs. In order to obtain 1.2 kW DC-DC full bridge converter gain values, the selected parameters values are: %*OS* = 0.1% and $T_s$ = 1.9 ms. For bidirectional power transfer between buses, the step-down version 3.3 kW 380 V to 190 V DC-DC full bridge converter takes the following parameters for desired polynomial are: %*OS* = 0.1% and $T_s$ = 1 ms. The step-up version 3.3 kW 190 V to 380 V uses next parameters for this desired polynomial %*OS* = 1% and $T_s$ = 2 ms. In all cases, auxiliar pole is placed five times real part of the complex conjugate value.

To obtain the gains for the full bridge DC–DC converters for the charge–discharge of the ESS in the primary bus; the first one for battery charge, 2.5 kW and 380 V to 250 V, uses the following parameters to define the desired polynomial: %*OS* = 1% and $T_s$ = 1.5 ms. On the other hand, the converter that manages the battery discharge, 2.5 V and 250 V to 380 V, uses the following parameters to define the desired polynomial: %*OS* = 1% and $T_s$ = 2.3 ms. Finally, for the charge–discharge of ESS in the secondary bus. The first converter that manages charge cycles, 1.25 kW and 190 V to 125 V, uses the following parameters to define the desired polynomial: %*OS* = 1% and $T_s$ = 1.1 ms. The converter that perform discharge events for batteries uses the following parameters for desired polynomial: %*OS* = 1% and $T_s$ = 1.6 ms. Again, the auxiliar pole is placed five times the real part of the complex conjugate of the dominant pair. Results apperar in Tables 8 and 9.

Behavior of DC–DC full bridge converter, all versions, performing in close loop and facing load changes are presented and commented in next section.

**Table 8.** Gain values for both vesions of Boost converter.

| Version | $K_1$ | $K_2$ | $K_3$ | $K_I$ |
|---|---|---|---|---|
| 380 V | 0.996579 | 0.205617 | 0.030030 | 12.038176 |
| 190 V | 0.623042 | 0.119862 | 0.005269 | 17.391827 |

**Table 9.** Gain values for all the vesions of DC–DC full bridge bridge converter.

| Version | $V_{in}$ | $V_{out}$ | $K_1$ | $K_2$ | $K_I$ |
|---------|----------|-----------|-------|-------|-------|
| 400 W | 190 V | 48 V | 0.065227 | 0.098410 | 364.003904 |
| 1.2 kW | 380 V | 190 V | 0.020290 | 0.039492 | 35.027654 |
| 3.3 kW | 380 V | 190 V | 0.032890 | 0.054304 | 117.792483 |
| 3.3 kW | 190 V | 380 V | 0.008089 | 0.027744 | 5.407997 |
| 2.5 kW | 380 V | 250 V | 0.013945 | 0.033055 | 31.541976 |
| 2.5 kW | 250 V | 380 V | 0.009073 | 0.032140 | 12.959751 |
| 1.25 kW | 190 V | 125 V | 0.029152 | 0.047667 | 83.196595 |
| 1.25 *KW* | 125 V | 190 V | 0.0018966 | 0.047627 | 39.884327 |

### 5.1.3. Cuk Converter Controller Development

For Cuk converter controller design, classical control techniques are used. The 400 W DC-DC full bridge converter version is the previous stage to this converter and sets 48 V as input for the driver, which energizes the 60 W LED luminary at 68 V. The block diagram of the controller is shown in Figure 8.

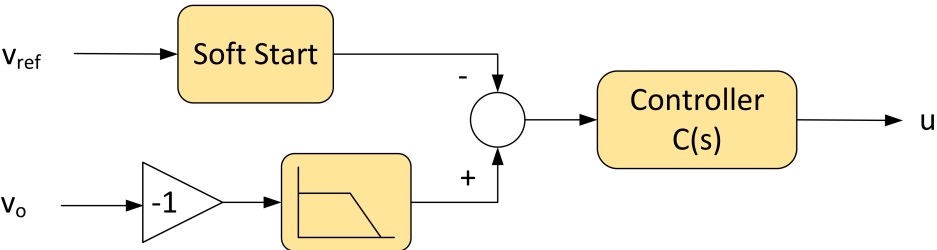

**Figure 8.** Cuk controller structure.

Due to the nature of the Cük converter, the output voltage is obtained with a negative polarity; for this reason, the measured voltage is multiplied by a gain value of −1. The measured voltage signal passes through a low-pass filter in order to make the control using the direct component of the converter output signal. On the other hand, the reference signal passes through a block that incorporates first order dynamics to achieve the effect of a soft start of the converter. Finally, the controller provides low-frequency gain, so it is proposed to place a zero at 400 Hz, so that the expression for the controller is given as:

$$C(s) = 1 + \frac{2\pi(400 \text{ Hz})}{s} \tag{9}$$

To validate the operation of the converter and the LED luminaire, the closed loop converter perform appears in results section.

### 5.1.4. DC–AC Bidirectional Full Bridge Converters Controllers Development

The controller for DC–AC bidirectional converter must be able to regulate DC bus and, at the same time, operates with unitary power factor, both tasks performing as rectifier. The other way, besides inject power to mains, converter is able to compensate reactive power and harmonics operating in inverter mode. Controller structure is composed by two stages, reference signal generation and voltage–current cascade controller, controller output is a set of sinusoidal signal *u* that is processed in a SPMW modulator to obtain a 3 *ϕ* switching pattern for converter power devices. The complete schematic is shown in Figure 9.

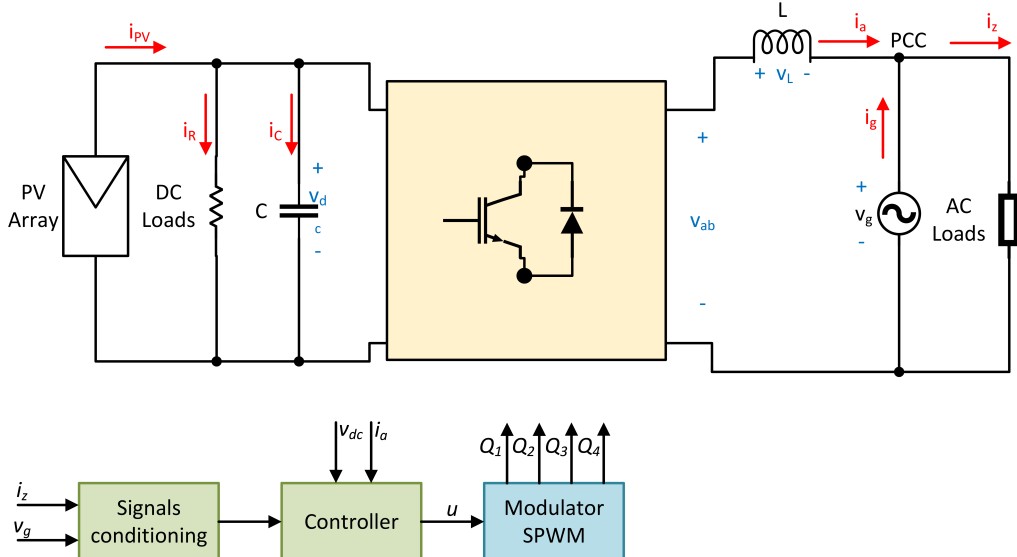

**Figure 9.** Converter and controller scheme.

Single phase *dq* transformation is applied to current circulating in AC side, $i_z$, to place the variable in a rotating reference frame (see Figure 10).

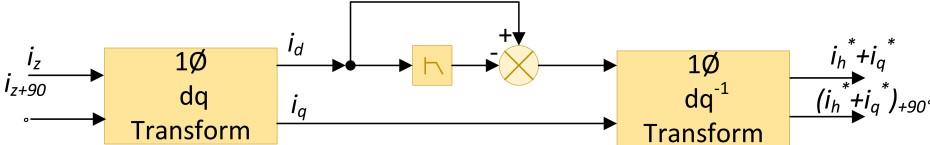

**Figure 10.** Reference signal generator.

The *dq* transform is applied to current circulating in AC side, $i_z$, to place it in a rotating reference frame, mathematically depicted as:

$$\begin{bmatrix} i_d \\ i_q \end{bmatrix} = \begin{bmatrix} sin\phi & -cos\phi \\ cos\phi & sin\phi \end{bmatrix} \begin{bmatrix} i_z \\ i_{z+90°} \end{bmatrix}$$

In this rotating frame $i_z$ current components can be separated in: $I_d$, term corresponds to active power and $I_q$ referred to reactive power; the sum of the remaining terms meet current harmonics components, $i_h$. After transformation, $i_d$ and $i_q$ are:

$$\begin{aligned} i_d &= I_d + \tilde{i}_d \\ i_q &= I_q + \tilde{i}_q \end{aligned} \tag{10}$$

Resultating signal are composed by a constant levels of $I_d$ and $I_q$ and several frequencies components, $\tilde{i}_d$ y $\tilde{i}_q$; it can de expressed as (10). In order to do nonactive power compensation, $I_d$ is filtered; this way, active power is separated. Then, the obtained signal is subtracted from $i_d$ and takes back to the time reference frame through *dq* inverse transform; this mapping operation is defined by:

$$\begin{bmatrix} i_d^* + i_h^* \\ (i_d^* + i_h^*)_{+90°} \end{bmatrix} = \begin{bmatrix} sen\phi & cos\phi \\ -cos\phi & sen\phi \end{bmatrix} \begin{bmatrix} i_d - I_d \\ i_q \end{bmatrix}$$

For this application, control objectives are: DC bus regulated and active power management. The control structure shown in Figure 11 includes two PI classical controllers in a cascade configuration. The external loop regulates DC bus to reference value $v_{dc}^*$, and its output is a part of current signal reference $i_d^*$ referred to active power demanded by load.

The complete expression for inner loop reference is the sum of $i_d^*$, non-active terms $i_q^* + i_h^*$ and a compensation term $\frac{2}{V_g}$.

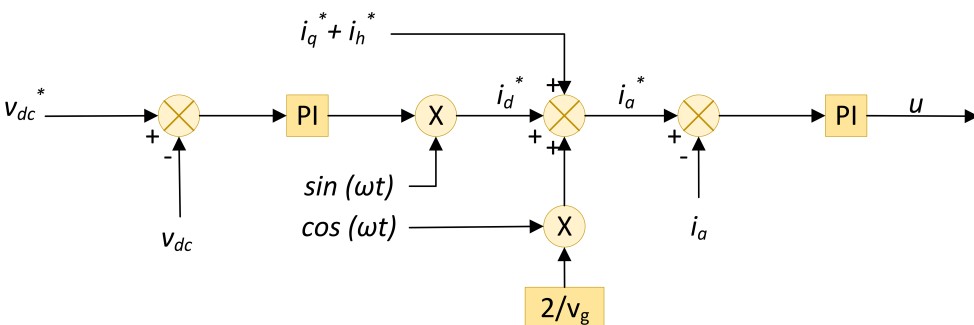

**Figure 11.** Cascade controller structure.

Considerations for controller design are: first, current inner loop is, at least, ten times faster than voltage external loop; second, for tuning, external loop PI controller gain must be selected carefully to avoid important current transients if an excessive value were selected. Parameter values for PI controllers for the 3.3 kW version at 220 $V_{RMS}$, and parameter values for PI controllers for the 1.5 kW version at 127 $V_{RMS}$ appear in Table 10.

**Table 10.** Parameter values for DC–AC full bridge PI controllers.

| Version | $K_v$ | $\tau_v$ | $K_i$ | $\tau_i$ |
|---------|-------|----------|-------|----------|
| 3.3 kW | 10 | 0.12 s | 0.08 | 0.03 s |
| 1.5 kW | 10 | 0.1 s | 0.08 | 0.025 s |

Behavior of DC–AC bidirectional single phase full bridge converter performing in close loop taking and injecting power to mains are presented and commented in next section.

### 5.2. Energy Management System Design

As supervisory control level, an EMS is included to control the multibus multivoltage DCMG. The main idea to design the EMS is to perform in a similar way to an industrial production system. The EMS is programmed in a central concentrator (CC) hardware; from here, enable signals are generated in order to harness all the energy harvested in PV arrays in various consumption points of the system. In order to reach harmonic performance of the DCMG, a list of loads priorities have to be met:

1.  To supply loads into the building (lighting system and electronics loads)
2.  To perform charge EV sequence or inject power to mains
3.  To carry out the charge of the ESS batteries

Once the order of priorities is known, initially, EMS performs a logic startup to define an initial condition of the sistem called initial state of the system ($\mu_{ko}$). A first sequential stage takes control of those PEC that can provide power to the system eventually, as a source condition. Another sequential stage controls the system load condition by enabling PECs that may consume power in certain periods of time. In order not to occupy too many resources of the CC, source condition and load condition changes are performed by one finite machine state (FMS) each.

The EMS design is based in Petri net theory. Petri nets applications are popular in industrial production items, logistics, resources management and others. In other hand, just a few developments are reported in power systems such as study of protection dynamic behavior [41] and hierarchical restoration after a fail event [42]. Nevertheless, due to discrete nature of the concept, it is possible to use a Petri net to read the new measurements, process them and update the system state of the corresponding transitions (enable signals)

in a fixed period of time. The basic fundamentals of Petri net appear in a summarized way in [43].

EMS proposed consists of three stages: the startup test, source converters perform in closed loop at one tenth of the rated power to know how many and what are the PECs available to work. With the information collected, it is possible to set the DCMG initial state $\mu_{ko}$. The second stage is a set of FMS to enable-disable source PECs; the most adequate FMS, in the current period, is selected to perform in concordance with the previous state of the system ($\mu_{k-1}$). In the same way, the third stage is a set of FMS to enable–disable load PECs, the most adequate FMS; for the current period, it is selected to perform in concordance with the previous state of the system. An EMS block scheme is shown in Figure 12.

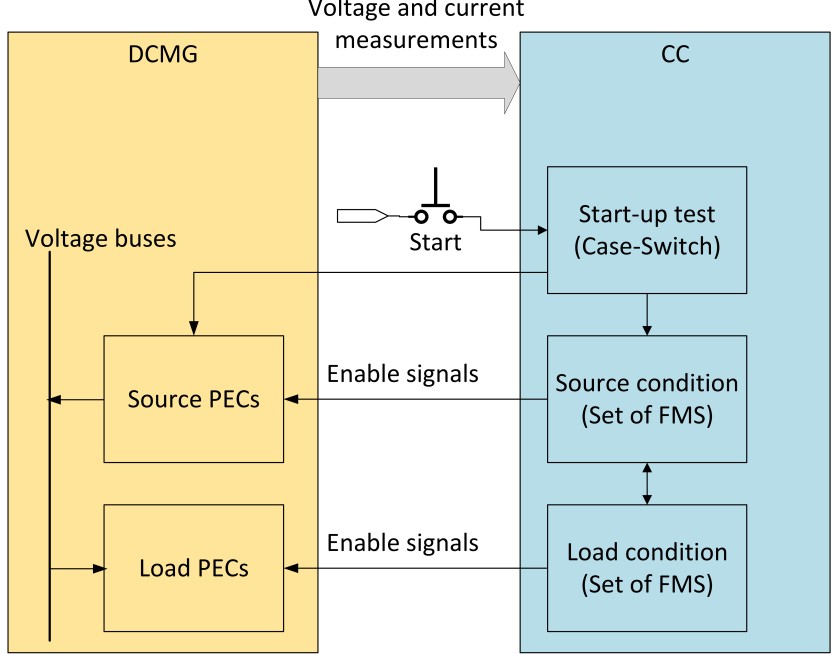

**Figure 12.** EMS stages block diagram.

After startup is performed and the initial state ($\mu_{ko}$) is defined once, then, the two FMSs are selected to move to a new states ($\mu_k$, $\mu_{k+1}$, . . . ). Voltage and current measurements are elements of the state vector, which is divided into two: the part associated with sources and the part associated with loads. In order to get the next period of time ($k + 1$), measurements are evaluated by the EMS and the FMSs activates the necessary enable signals (transitions) to move to a new state or remains in the same state for another period of time. The DCMG as a Petri net appers in Figure 13.

In Petri nets theory, four basic elements can be identify: place (circle), transition (rectangle), arc (arrow) and token (dot), they are associated with DCMG elements in a specific way. A place can represent as one of the following options: a tandem boost PV array and boost PEC ($APV1M$–$APV4M$), a DC-DC full bridge PEC and the electronic load ($EC$) or six LED luminaries ($LED$), ESSs battery charger PEC ($BAT1C$–$BAT2C$), also primary bus and secondary bus are considered ($BUSP$, $BUSS$) places, but they are not able to store power more than one period of time. A transition represents the enable signal of a PEC, since Petri net point of view, an active transition allows the power flow ($T0$–$T17$). An arc is the medium through power flows, it also indicates the direction of the flow. A token represents a power unit (W, kW, etc.). The places $APV1M$–$APVM4$ are connected to the respective bus through two trajectories arc and transition. One transition is activated if boost converter operates with feedback state controller in closed loop, voltage regulation mode. The other transition is turned on when this kind of PEC operates in MPPT mode.

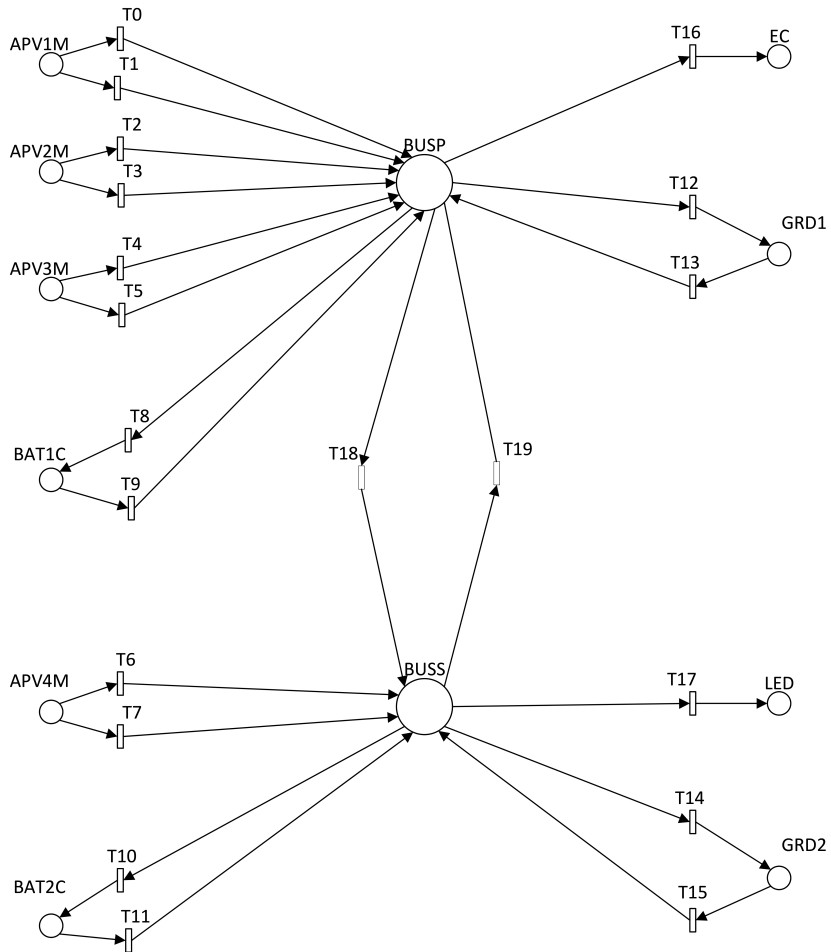

**Figure 13.** DCMG multibus multi-voltage Petri net scheme.

CC receives and quantifies the following measurements: PV arrays power output, the three linked to primary bus ($PV1$–$PV3$) and for the PV array related to secondary bus, $PV4$. Generally, if $i = 1, 2, 3, 4$ then $PVi$ is any of the PV arrays. If $PVi = 0$ the PV array is not working because is not connected or damaged. If $PV1 = 1$, low power is harvested form PV array, enough to set the bus, but not a lot of loads can be supplied. At $PVi = 2$, there is power enough to supply all the loads in the building and inject power to mains. If $PVi = 3$ there is surplus power harvested, tasks before mentioned can be done also a ESS batteries charge sequence is possible. This quantification is process in CC in binary form using two digits.

In a similar way, ESS batteries SoC is associated with parameters $BAT1$, for ESS in primary bus and, $BAT2$ is relaed to secondary bus. In general way, if $i = 1, 2$ then $BATi$ is any of the ESS batteries. Now, if $BATi = 0$ ESS is not working for one of the following reasons: ESS is not connected or ESS is damaged. If $BAT = 1$, SoC is low and is not possible to keep the the bus regulated with some loads. When $BAT1 = 2$, SoC is at medium level is possible to supply loads and perform a charge sequence for an EV. If $BAT = 3$, SoC level value is high, then, all the loads into the building can be supplied, perform charge sequence for EV or inject power to mains. Again, $BATi$ value es represented with two binary digits.

Mains has two point of connection to DCMG, one through primary bus, $GRD1$, and the other via secondary bus, $GRD2$. Again, to assert in general way, $i = 1, 2$, then $GRDi$ is any of mains voltage measurements. The quantification associated to these parameters is: if $GRDi = 0$ there is not voltage in mains, else, if $GRDi = 1$, there is voltage in point connection with mains. At this time, $GRDi$ is a binary number with two possible values.

All the paramemters before mentioned are concatenated to build the DCMG state vector. Depending on the state vector information, the needed transitions are activated to

move to a more convenient state in the corresponding FMSs to reach a new source condition or load condition or maintain the current state.

### 5.2.1. Startup Stage

In the proposed CDMG there are eight PEC able to supply system loads, source converters. The startup test stage enables all the source PECs at one tenth of rating power in closed loop and in stand-alone operation. The aim of this test is to verify how many and what PECs are able to be connected to the corresponding bus through plug-and-play. Once the PEC is enabled, the test last a period of time enough to let pass the transient and reach steady state, in this condition output voltage measuremet is compared with output reference value in order to verify if measurement value is into $\pm10\%$ reference value. If the test is pass by a specific PEC, the corresponding bit is set '1', else is set '0', and data is sent to CC. With the resulting eight bits information, CC takes a decision and set the system initial state of source condition. Let $SC$ be the source active bit vector defined and ordained as shown in Table 11.

**Table 11.** $SC$ vector.

| $SGD2$ | $SGD1$ | $SBT2$ | $SBT1$ | $SPV4$ | $SPV3$ | $SPV2$ | $SPV1$ |
|--------|--------|--------|--------|--------|--------|--------|--------|
| $Bit7$ MSB | $Bit6$ | $Bit5$ | $Bit4$ | $Bit3$ | $Bit2$ | $Bit1$ | $Bit0$ LSB |

In a detailed way, $SGD1$ and $SGD2$ are bits associated with the DC-AC full bridge converters condition, particularly when perform in rectifier mode as source PEC. If one of them passes the initial test or stays in on mode then $SGDi = 1$, else, $SGDi = 0$ if not passes the initial test or stays in off mode. $SBT1$ and $SBT2$ are bits associated with the step-up DC-DC full bridge converters at the ESSs, in batteries discharge mode. If one of them passes the inicial test or stays on mode $SBTi = 1$, else, $SBTi = 0$ if not passes the initial test or stays in off mode. Finally, $SPV1$, $SPV2$, $SPV3$ and $SPV4$ are bits associated with the boost converters condition, specially when performs in regulation mode. If $SPVi = 1$, PEC is able to operate or stays in on mode, else, $SPVi = 0$ if not able to operate or in off mode.

With eight PECs in the system, one bit each, there are 256 possible options to initialize the DCMG operation. In the start-up stage, the CC prioritizes the consumption of power collected in PV arrays, then ESS power is harness, finally, power from mains is taken. It means, only 22 combinations are available options to become initial DCMG state, most of the using PV arrays. If there is no PEC able to pass the start-up test, DCMG goes back to system off condition.

### 5.2.2. Source Condition Establishment Stage

After the startup test, the initial state is set in the DCMG already, a FMS is selected from a set. There are some options for FMS, depending on state vector $v_k$ measurements. If $V_k$ shows another sosource ready to provede power to system, vector $SC$ change its value in the corresponding parameter, a new FMS is selected.

### 5.2.3. Load Condition Establishment Stage

In the proposed DCMG, there are six load PECs. The most critical loads are lighting system (LLED) and DC-DC full bridge converters to supply electronic loads (LEC). The rest of the load PECs are: two step-down DC-DC full bridge converters to charge batteries in ESSs and the before mentioned DC-AC full bridge converters, particularly in inverter mode. If the load PEC is demanding power, its bit value is set '1', else is set '0', and data is sent to CC. With the resulting six bits information, CC takes a decision and set the state of the FMS load condition, FMS source condition state is already known. Let be $LC$ the load active bit vector defined and ordained as shown in Table 12.

**Table 12.** *LC* vector.

| *LGD2* | *LGD1* | *LBT2* | *LBT1* | *LLED* | *EC* |
|---|---|---|---|---|---|
| B5 | B4 | B3 | B2 | B1 | B0 |
| MSB | | | | | LSB |

In a detailed way, $LGD1$ and $LGD2$ are bits associated with the DC–AC full bridge converters condition when perform in inverter mode, as load PEC. If one of them is injecting power to mains $LGDi = 1$, else, $LGDi = 0$ if is not injecting to mains or stays in off mode. $SBT1$ and $SBT2$ are bits associated with the step-up DC–DC full bridge converters at the ESSs, in batteries charge mode. If one of them is in charging sequence $LBTi = 1$, else, $LBTi = 0$ if stays in off mode. Finally, $LLED$ and $LEC$ are bits associated with the DC–DC full bridge converters, the first one supplies one to six cuk converter and LLED luminarie and the second powers electronic loads in a teachers office. Both, If $LLED = 1$ is suppling to lighting system, else, $LLED = 0$ if PEc is in off mode. Same logic is applied to $LEC$.

## 6. Results

In order to validate the novel architecture concept Psim simulation results are presented. The section is divided into two subsections: first, simulation results of converters performing in closed loop in single stage. Second, some trajectories of the multibus multi-voltage DCMG architecture supplied form PV arrays are tested to validate the correct operation of the system only with local control.

### 6.1. Performance of Converter in Stand-Alone

In order to validate models, designs and controllers of each converter version is presented performing in stand-alone, most of cases supplied from an ideal source to validate stand alone performance while faces load changes. Inductor current ripple criterion must be met to avoid high frequency components circulation during cascade connection events.

### 6.1.1. Boost Converter Closed Loop Performance

Boost converters in the DCMG are able to operate in two modes, output voltage regulation using and MPPT. Firstly, the both versions of boost converter in regulation mode behavior are verified with the same load sequence. The simulation test starts at 100% load condition, then, 22.5% load steps are removed until reach a final load at 10%. At 0.5 s, 22.5% load steps are connected back until meet 100% load condition again. First version, 380 V output voltage boost converter behavior appears in Figure 14.

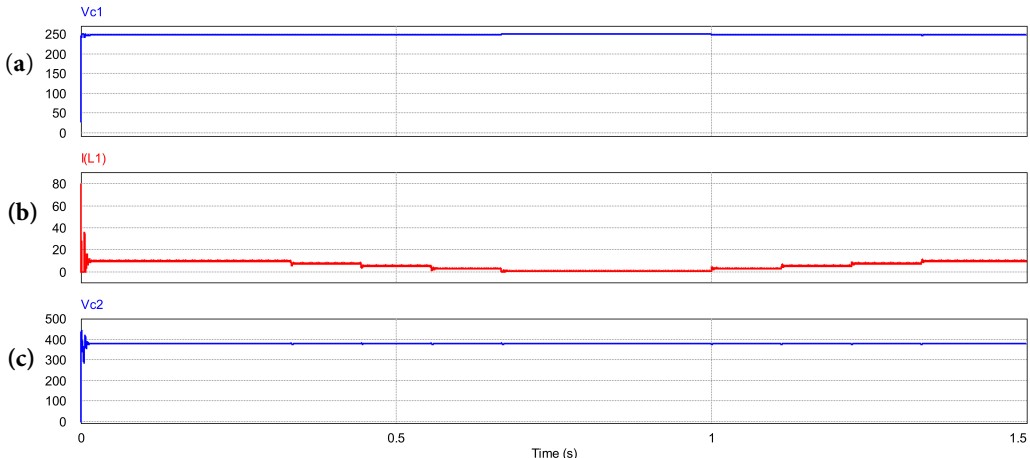

**Figure 14.** Boost converter 2.5 kW at 380 V with state feedback controller. (**a**) input voltage, (**b**) inductor current and (**c**) output voltage.

No significant dynamics is observed in capacitor input voltage at starting or during load steps, see Figure 14a. Inductor current ripple is not greater than 10% nominal value and converter performs in CCM all the simulation time, see Figure 14b. Overshot is noticed in DC bus voltage at the starting because not higher level of control is included, in steady state, transitions are smooth during load changes, see Figure 14c. Boost converter at 190 V output voltage simulation results appears in Figure 15.

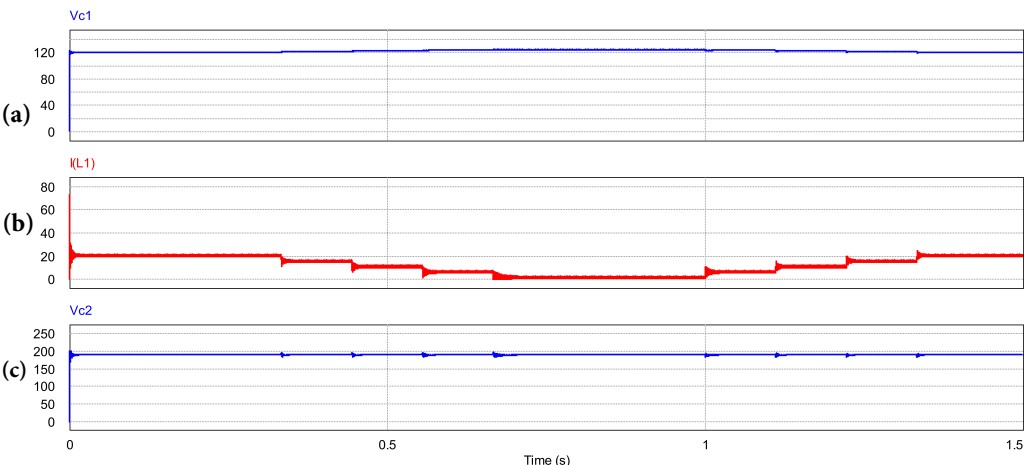

**Figure 15.** Boost converter 2.5 kW at 190 V with state feedback controller. (**a**) input voltage, (**b**) inductor current and (**c**) output voltage.

Again, no important transients are observed in capacitor input voltaje at starting neither during load transitions, see Figure 15a. Inductor current ripple keeps in criteria value and the converter performs in CCM, see Figure 15b. Starting oscillations take more time than those in 380 V version, they appear at starting and load transitions but overshot is not significant, see Figure 15c. A simulation to evaluateas a 380 V boost converter in MPPT mode is perform using *P&O* algorithm, Figure 16.

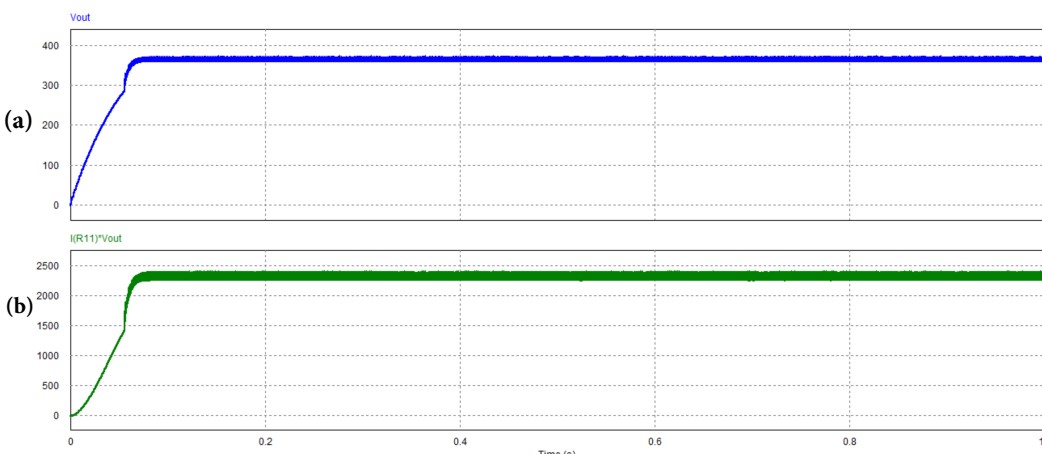

**Figure 16.** Sixth set of simulation results, (**a**) primary DC bus and (**b**) power harvested from PV array.

A sequence from 0% to 100% PV array rating power shows: in Figure 16a boost converter output voltage to set primary bus, mainly voltage value keeps around 380 V but, voltage ripple performing in this mode is larger than in regulation mode deu to. In Figure 16b waveform of power taaken form PV array in time domain.

### 6.1.2. DC-DC Full Bridge Closed Loop Performance

In the DCMG proposed there are several DC-DC full bridge converters versions, in this subsection some of this versions results are presented, PECs selected to test are 400 W at 48 V, 1.2 kW at 190 V and 3.3 kW at 190 V. First, 400 W at 48 V topology supplies six 60 W tandems Cuk driver-LED luminary and the simulation is relative to this dynamics. Simulation results are shown in Figure 17.

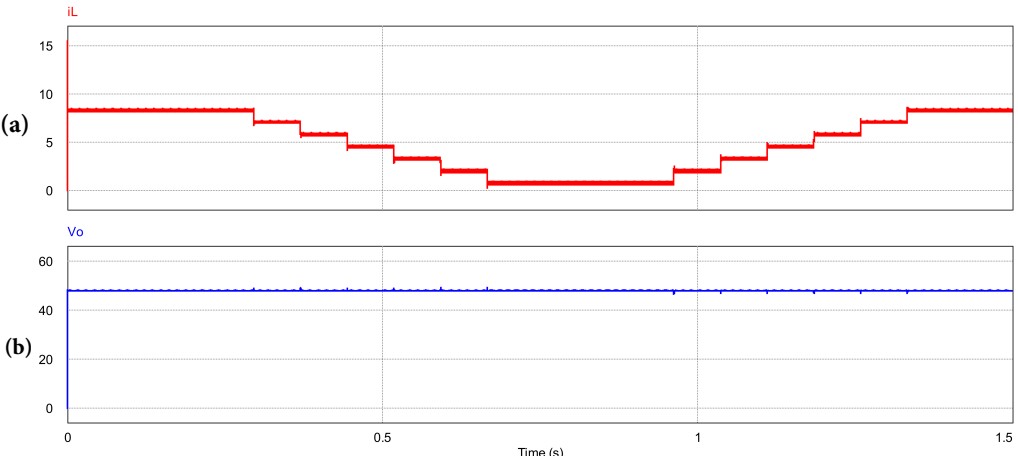

**Figure 17.** DC-DC full bridge converter 400 W at 48 V with state feedback controller. (**a**) inductor current, (**b**) output voltage.

Simulation test starts with 100% load condition, then tandems are disconnected one by one until converter remains at 10% load operation, only with bleeding resistor connected. Further, tandems are connected one by one to meet 100% load condition. In this case, inductor current and output current is the same. Inductor current ripple keeps its value according the critieria and converter keeps on CCM during the test, Figure 17a. There are not significant overshot in DC bus at luminaries connection/disconnection sequence, Figure 17b. Converter DC-DC full bridge 1.2 kW at 190 V output version behavior is tested, see Figure 18.

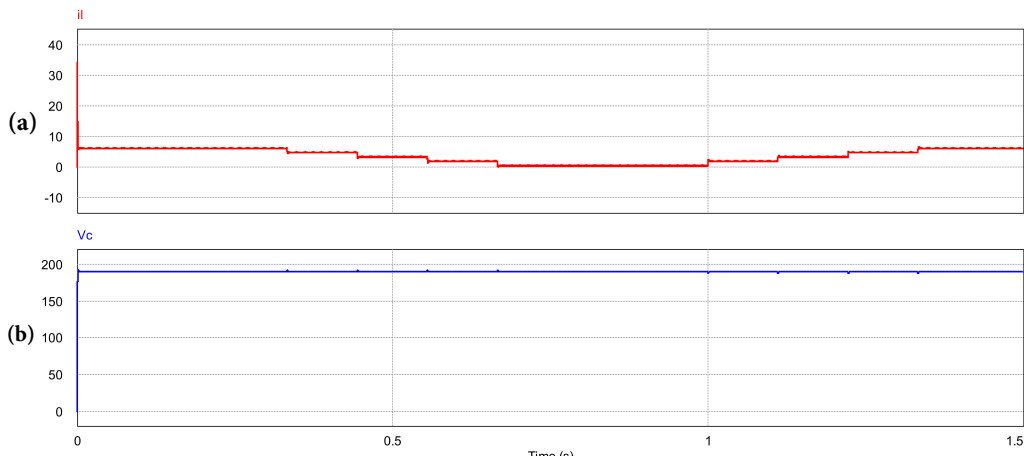

**Figure 18.** DC-DC full bridge converter 1.2 kW at 190 V with state feedback controller. (**a**) inductor current, (**b**) output voltage.

Again, simulation starts at 100% load condition, then 22.5% load steps are removed until 10% nominal load. At 1 s, 22.5%, load steps are connected back until meet 100% load condition again. Smooth transients presence is noticed in inductor current ripple and CCM condition remains during the test, see Figure 18a. No significant overshot appears in DC

bus at load connection/disconnection sequence, Figure 18b. Finally , converter DC-DC full bridge 3.3 kW at 190  V version behavior is tested in the same way than before mentioned versions, see Figure 19.

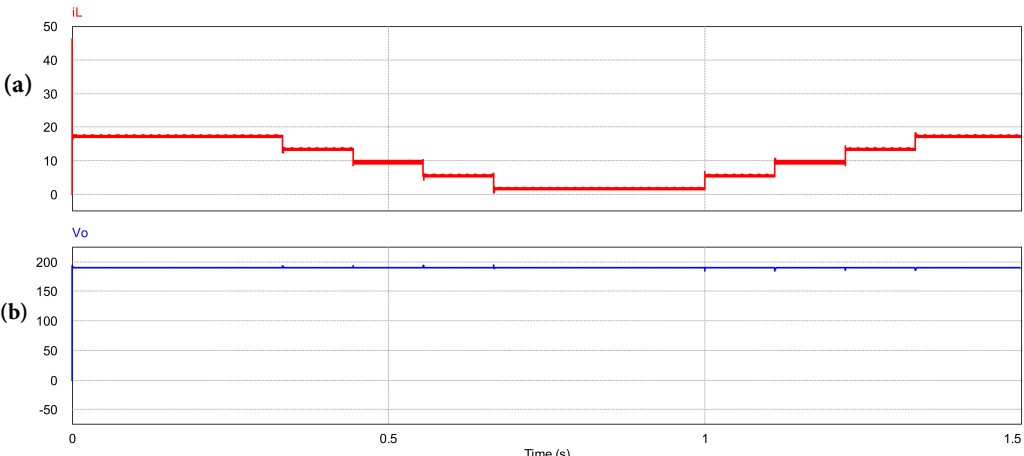

**Figure 19.** DC-DC full bridge converter 3.3 kW at 190 V with state feedback controller. (**a**) inductor current, (**b**) output voltage.

Test simulation starts at 100% load condition, then 22.5% load steps are disconnected until final load at 10% nominal power. At 1 s, 22.5% load steps are connected back until meet 100% load condition again. Current ripple increases at 55% load condition but ripple current criteria is met and CCM reamins during the test, see Figure 19a. Smooth transients apear in DC bus, with no overshot, at load connection/disconnection sequence, see Figure 19b. All the DC-DC full bridge converters versions remaining are tested in the same way, similar results were obtained in all cases.

### 6.1.3. Cük Converter

Cük converter closed loop behavior is validated with a simulations. Simulation time goes from soft start to steady state. No load changes are considered because a PV array with no variant features is always supplied by the driver. Four dynamics appear in model in (3), inductor one current and capacitor one voltage are shown in Figure 20.

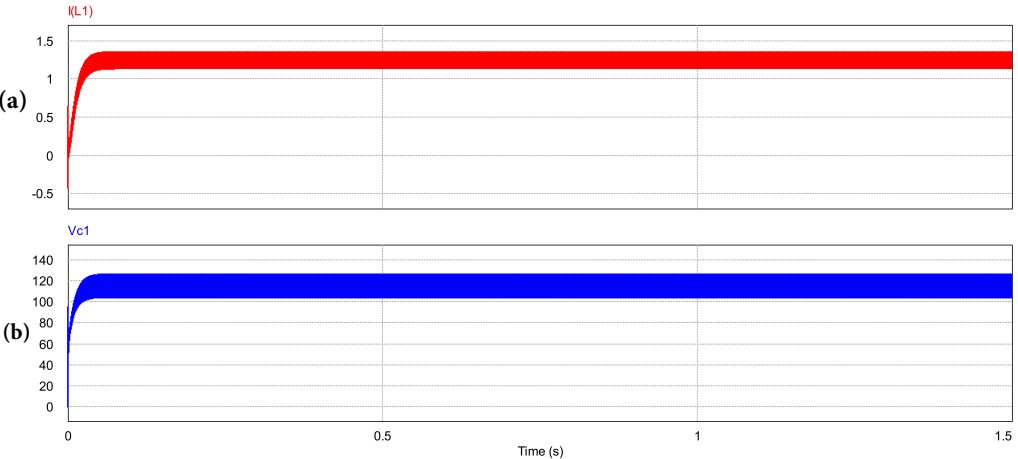

**Figure 20.** Cuk converter performing in closed loop. (**a**) inductor one current, (**b**) capacitor one voltage.

A 0.05 s soft start is perform at the beginning, no transient and well regulated variables are observed. The other pair of state variables, inductor two current and capacitor two voltage appear in Figure 21.

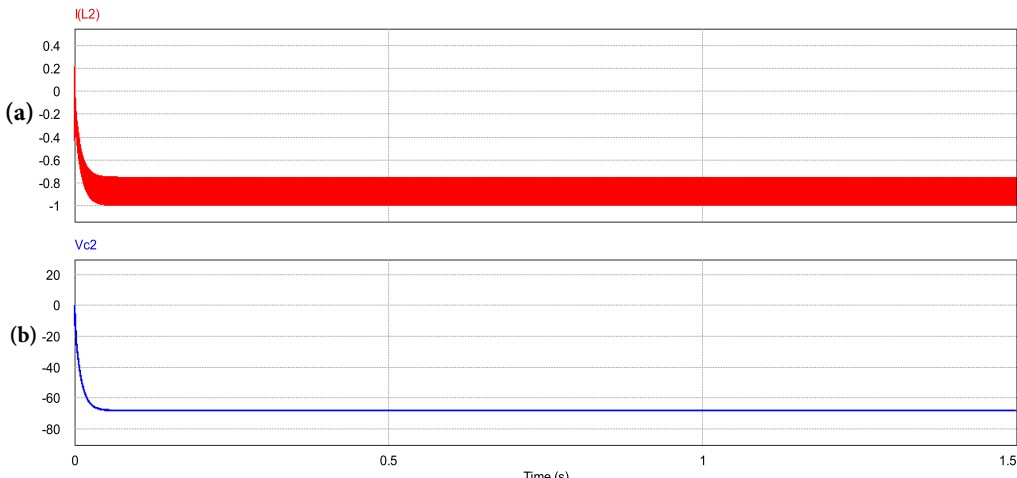

**Figure 21.** Cuk converter performing in closed loop. (**a**) inductor two current, (**b**) capacitor two voltage.

Inductor two curren ripple is 20% nominal value, this condition avoids undesirable effects in light spectrum. Voltage output, $v_{C2}$, performs a smooth start and reminds regulated during simulation time.

### 6.1.4. DC-AC Bidirectional Single Phase Full Bridge Converter

This converter features bidirectional capability, then, two simulations are performed in order to vaulate DC-AC single phase full bridge converter. The first one, in rectifier mode, 1.5 kW are transferred from AC to DC side, Figure 22 shows behavior in steeady state, negative sign in active power indicates power flows from the grid to DC side. Regulated DC bus with small voltage ripple can be seen in Figure 22a, mains current and voltage appear in Figure 22b, a 180° angle phase exists between grid voltage and grid current, this condition define the sign for active power; active and reactive power appears in Figure 22c reactive power is near to zero and converter operates with unitary power factor.

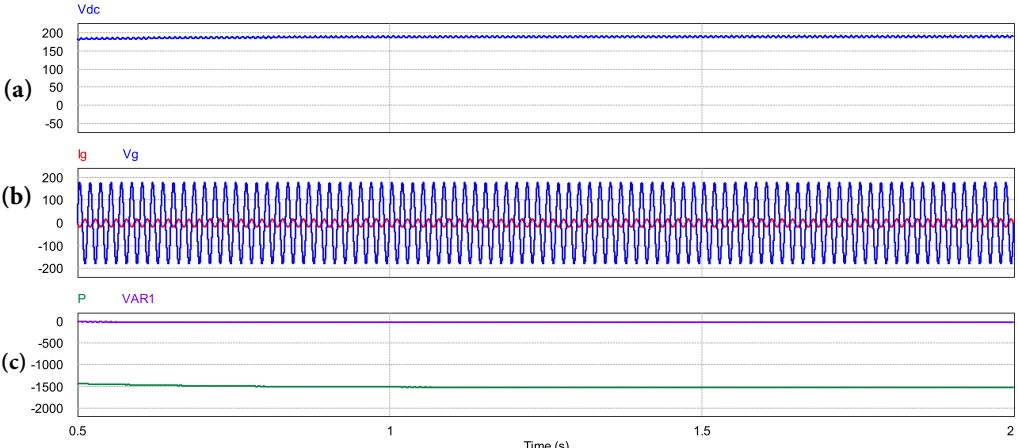

**Figure 22.** 1.5 kW DC-AC single phase full bridge converter, closed loop performing in rectifier mode. (**a**) DC bus voltage, (**b**) grid current *(red)* and grid voltage *(blue)*, (**c**) active *(green)* and reactive *(purple)* power.

Next simulaton corresponds to inverter operation, 1.5 kW are transfered from PV array to mains, Figure 23 shows behavior in stady state, positive sign in active power

indicates power flows from DC side to the grid. Regulated DC bus with small voltage ripple can be seen in Figure 23a, mains current and voltage appear in Figure 23b, there is a zero phase angle between gird current and voltge, then, the positive sign is defined for active power, active and reactive power appears in Figure 23c; and there is no reactive power to compensate.

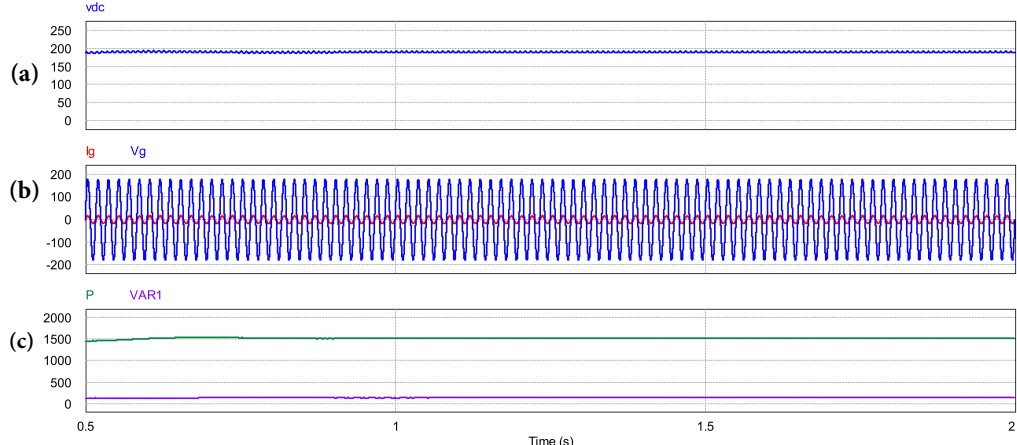

**Figure 23.** 1.5 kW DC-AC single phase full bridge converter, closed loop performing in inverter mode. (**a**) DC bus voltage, (**b**) grid current *(red)* and grid voltage *(blue)*, (**c**) active *(green)* and reactive *(purple)* power.

The second version in rectifier mode, 3.3 kW are transferred to DC side, Figure 24 shows behavior in steady state, negative sign in active power indicates power flows from the grid to DC side. Regulated DC bus with small voltage ripple can be seen in Figure 24a, mains current and voltage appear in Figure 24b, again a 180° angle phase exists between grid voltage and grid current, this condition define the sign for active power; active and reactive power appears in Figure 24c reactive power is near to cero and converter operates with unitary power factor.

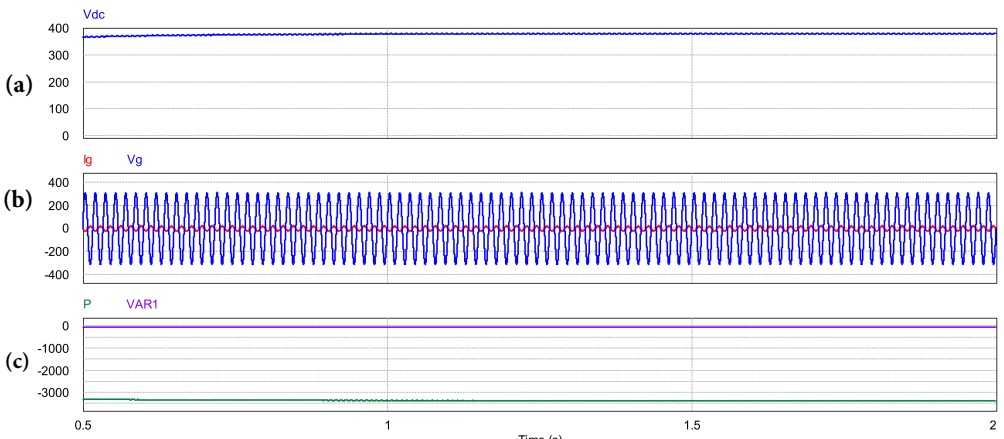

**Figure 24.** 3.3 kW DC-AC single phase full bridge converter, closed loop performing in rectifier mode. (**a**) DC bus voltage, (**b**) grid current *(red)* and grid voltage *(blue)*, (**c**) active *(green)* and reactive *(purple)* power.

This simulation corresponds to inverter operation, 3.3 KW are transfered from PV array to mains, Figure 25 shows behavior in stady state, positive sign in active power indicates power flows from DC side to the grid. Regulated DC bus with small voltage ripple can be seen in Figure 25a, mains current and voltage appear in Figure 25b, there is a zero phase angle between grid current and voltage, then, the positive sign is defined

for active power, active and reactive power appears in Figure 25c; and there is no reactive power compensation.

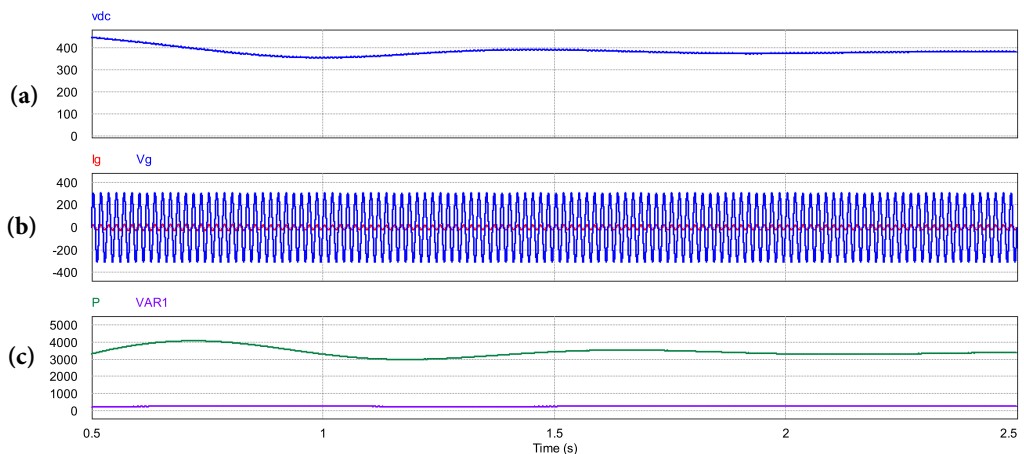

**Figure 25.** 3.3 kW DC-AC single phase full bridge converter, closed loop performing in inverter mode. (**a**) DC bus voltage, (**b**) grid current *(red)* and grid voltage *(blue)*, (**c**) active *(green)* and reactive *(purple)* power.

DC-AC bidirectional converter results confirm that is possible to have an DCMG able to interact with mains, it feature gives several possibilities for energy administration in the system.

### 6.2. Performance of DCMG Multibus Muti-Voltage

In order to validate multibus multi-voltage DCMG architecture concept two trajectories in the architecture are tested by simulation. Only state feedback controller as local control stage is included, no higher level controllers are considered. PECs are enabled by an digital signal and the output voltage reference signal. First trajectory to evaluate appear in Figure 26.

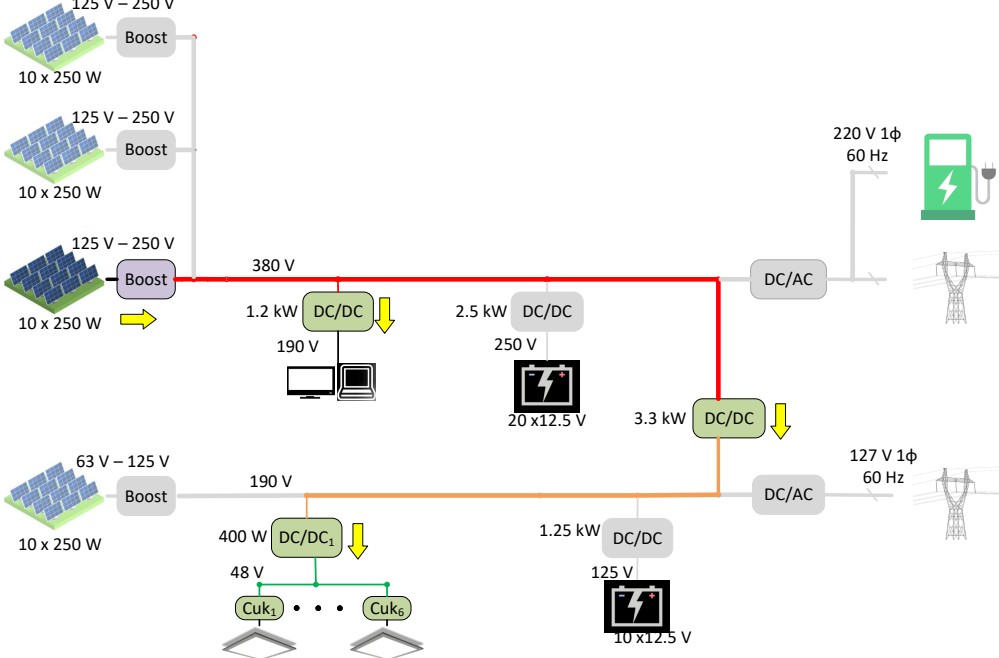

**Figure 26.** First test trajectory for DCMG multibus multi-voltage operating with only one boost converter.

Primary DC bus is generated by a single boost converter and supplies the following elements: a single 1.2 kW converter that supplies a 100 W electronic load; a 3.3 kW converter

to generate secondary DC bus that supplies a 400 W converter to supply six tandems Cuk driver-LED luminaire as final load. Simulation results are shown in Figure 27.

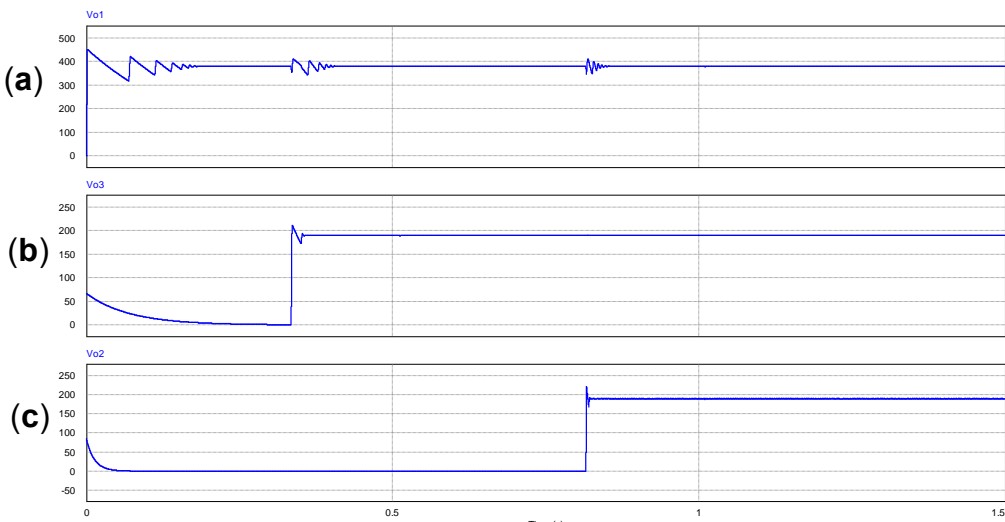

**Figure 27.** First set of simulation results, (**a**) primary DC bus, (**b**) 1.2 KW converter output voltage and (**c**) secondary DC bus.

Significant oscillations appear at the beginning of the simulation in primary bus, see Figure 27a; this kind of transient can be overcome using a secondary control loop in a centralized communication configuration for DCMG. Then, reduced amplitud oscillations are produced in primary bus by input capacitors in load converters being connected to primary or secondary buses. At 0.3 s, 1.2 kW converter is enabled and a smooth transient is produced in the primary bus Figure 27b. At 0.8 s secondary bus is established by 3.3 kW converter and corresponding transitory can be noticed in primary bus, again a negative exponential voltage dynamics can be noticed at the starting of simulation, see Figure 27c. Some results of the same test are presented in Figure 28.

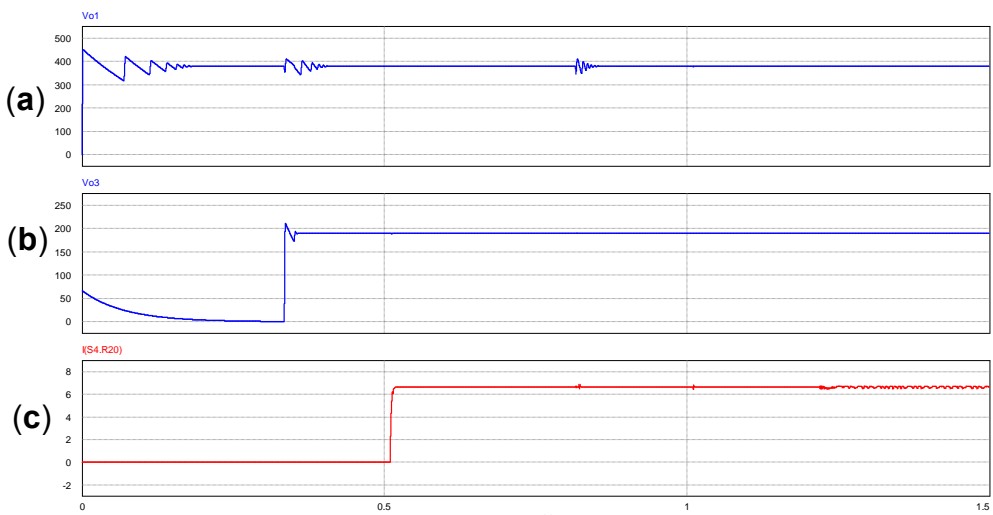

**Figure 28.** Second set of simulation results, (**a**) primary DC bus, (**b**) 1.2 kW converter output voltage and (**c**) electronic load input current.

Once 1.2 kW is enabled at 0.3 s, further at 0.5 s a personal computer is turned on, Figure 28. Waveforms are: in Figure 28a, principal bus voltage and Figure 28b, 1.2 kW converter output voltage are used like time references; Figure 28c shows current input

electronic load enabled at 0.5 s, some ripple can be seen caused by loads dynamic in secondary bus. More results are presented in Figure 29.

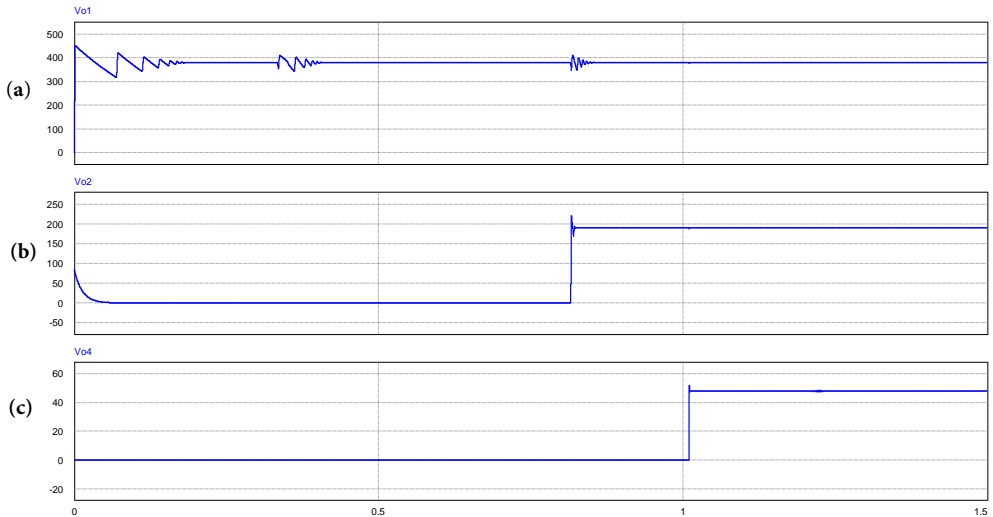

**Figure 29.** Third set of simulation results, (**a**) primary DC bus, (**b**) 1.2 kW converter output voltage and (**c**) terciary lighting bus.

Waveforms are: Figure 29a, primary bus voltage; and Figure 29b, 1.2 kW converter output voltage are used like time refrences; Figure 29c represents a tertiary lighting bus enabled at 1 s by 400 W converter, well regulated 48 V bus can be seen despite connection/disconnection of tandem driver Cuk-LED luminaries. The furthest functional trajectory for DCMG is from boost converter to lighting tandem driver-luminary, Figure 30 shows this power path.

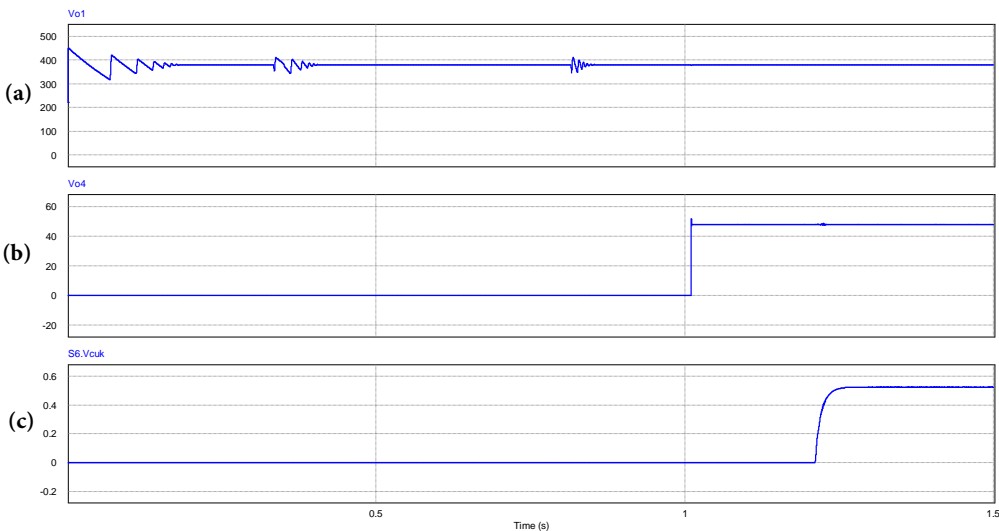

**Figure 30.** Fourth set of simulation results, (**a**) primary DC bus, (**b**) 1.2 kW converter output voltage and (**c**) Cuk driver reference signal.

Waveforms are: in Figure 30a, primary bus voltage as time reference; 400 W voltage output converter, Figure 30b, finally, Cuk convertert voltage output to supply one of the six luminaries on the set, see Figure 30c. The same longest trajectory is reviewed again, but luminaries connection appear in Figure 31.

In Figure 31a, primary bus voltage as time reference is shown; secondary bus is shown in Figure 31b, in Figure 31c, 400 W converter current output suppling six tandem Cuk

driver-LED luminary connection sequence, current ripple increases when another luminary is connected, at full load condition current ripple is about 10% of nominal inductor current.

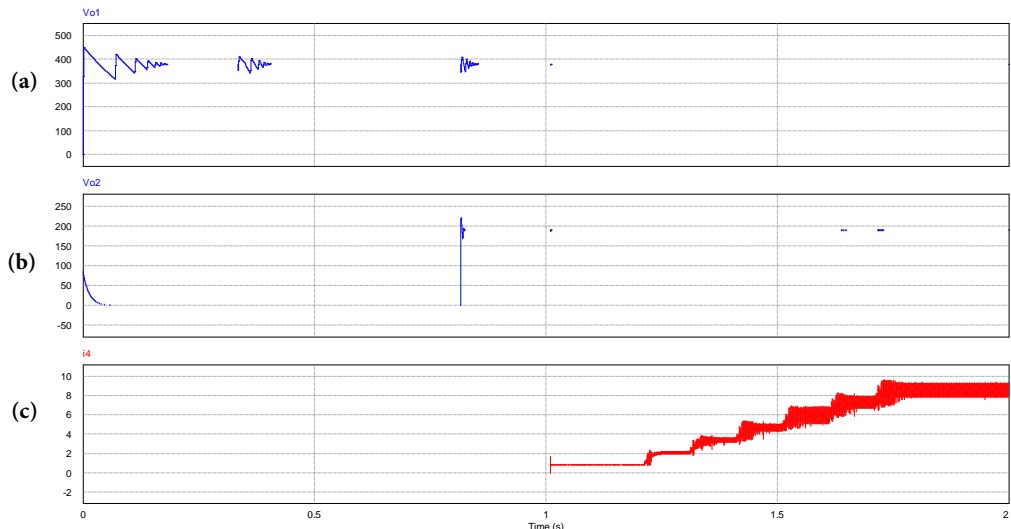

**Figure 31.** Fifth set of simulation results, (**a**) primary DC bus, (**b**) 1.2 kW converter output voltage and (**c**) 400 W converter current output.

A second test is done, now all three boost converters generate the primary bus, but the rest of the converters in the DCMG remain the same, again the aim is to supply an electronic load in primary bus and group of six tandem driver-luminary from secondary bus. An scheme of the new trajectory is presented in Figure 32.

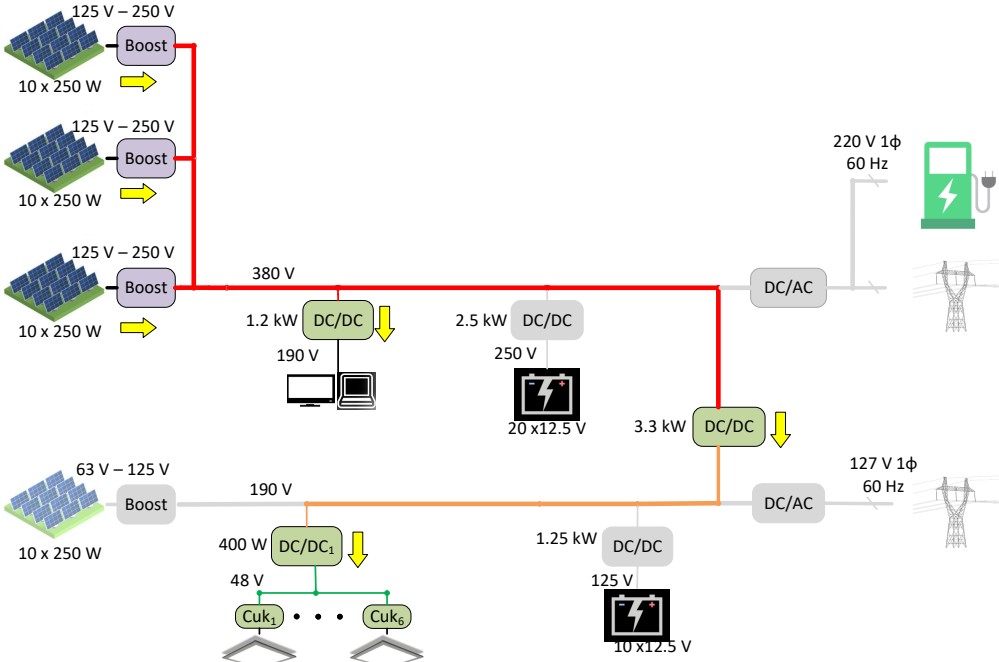

**Figure 32.** Second test trajectory for DCMG multibus multi-voltage performing with three boost converters.

In this case, it is necessary to verify current sharing using feedback state as local control strategy. The most important differences against test one appear in Figure 33, primary bus voltage presents starting transient with longer settle time than test one, see Figure 33a. Figure 33b shows total current from three boost converter and it is important to realize

current in Figure 33c, one boost converter current output is same waveform that total current but divided by three.

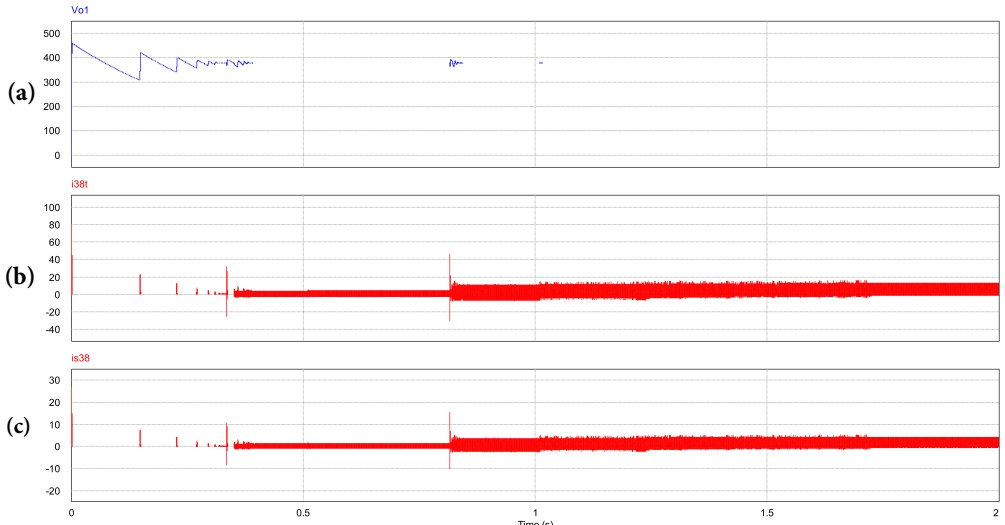

**Figure 33.** Sixth set of simulation results, (**a**) primary DC bus, (**b**) total current from three boost converters array and (**c**) boost converter number one output current.

### 6.3. Performance of DCMG with EMS

The EMS start-up stage let a pre-charge in the boost converter input capacitor then closed loop operation at one tenth of PEC rating power is perform, a simulation of this sequence using one of the 380 V output boost is shown in Figure 34.

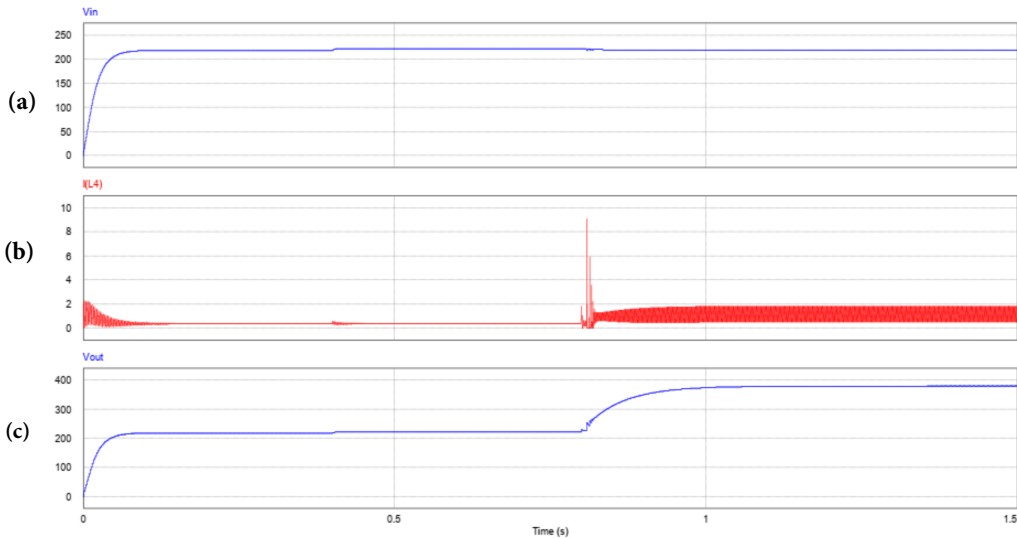

**Figure 34.** Start-up sequence with one 380 V output boost convertert, (**a**) input capacitor voltage, (**b**) inductor current (**c**) boost converter voltage output.

In Figure 34a input capacitor is charge through the thermistor at low temperature resistance value (10 Ω), at 0.4 s thermistor reduce its resistance value by temperature increase (0.22 Ω). Figure 34b, at the begining a transient current flows through inductor to charge output capacitor, at 0.8 s a second transient take place because the enable signal is set. In Figure 34c since the simulation starts, voltage output has the same value than input voltage by converter particular behavior. When converter is enabled reference value moves smoothly from pre-charge value to rating value, 0.8 s–1 s.

The star-up test stage considers eight source PECs, then 256 possible combination exist. Nevertheless, EMS prioritizes the usage of harvested power on PV array, then, as many times as possible, DCMG starts by using power collected in PV arrays through boost converters. Next simulation uses an incremental counter as input to EMS start-up to get the adequate signal output after the test, see Figure 35.

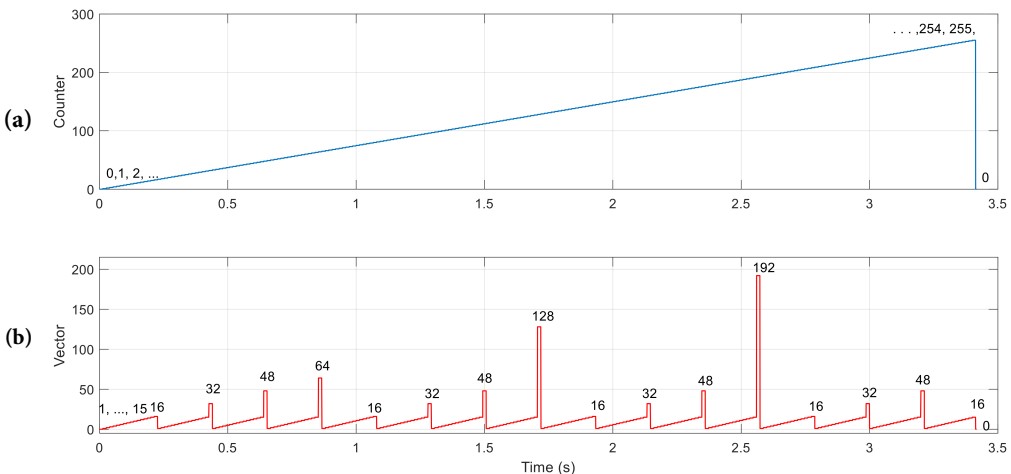

**Figure 35.** Sixth set of simulation results, (**a**) Incremental counter input and (**b**) start-up test ouput value.

According to *SC* vector, in Table 12, given values from 1 to 15 represents system start with PV arrays. If PV arrays do not pass the test, algorithm assumes as result one of the following: 16 (primary bus ESS batteries), 32 (secondary bus ESS batteries), 48 (both buses ESSs), 64 (power incoming from mains through primary bus), 128 (power incoming from mains through secondary bus), 192 (power incomng from mains through both buses). If there is no source able to supply power to the system output value is 0, it means, DCMG remains in off state and the building should be powered in a typical way, as an ultimate emergency action. A complete start-up test sequence is shown in Figure 36.

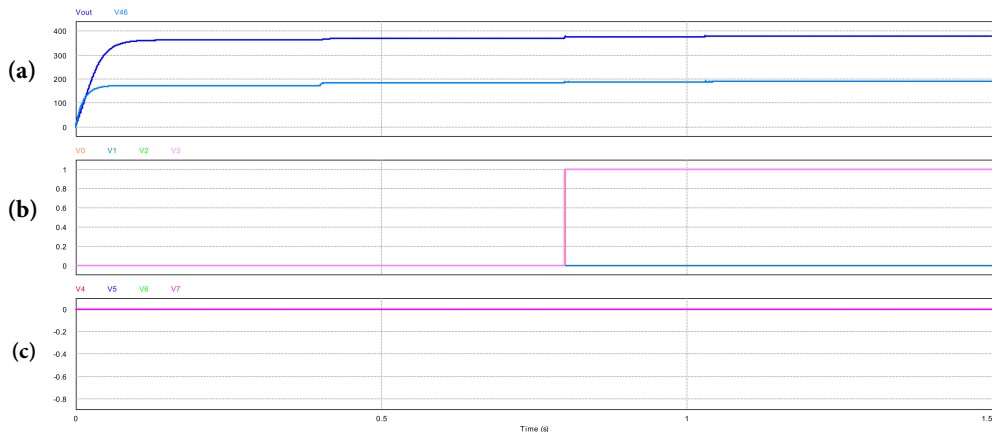

**Figure 36.** Sixth set of simulation results, (**a**) 380 V boost converter (*SPV*1, blue) and 190 V boost converter (*SPV*2, cyan), (**b**) *SC* vector, the least significant four bits (**c**) *SC* vector, the most significant four bits.

In Figure 36a, primary and secondary buses are established by one boost converter each, it is assumed that only this two PECs passed the initial test. In Figure 36b, $v_0 = SPV1$ and $v_3 = SPV4$ takes '1' value and the other two least significant bit renains in '0', finally, in Figure 36c the most significant four bits, related to ESSs batteries and DC-AC converters keeps on '0'.

## 7. Conclusions

In this paper a novel concept for DCMG architecture classified as multibus multi-voltage is presented, this development is suitable for building already in operation where different kinds of loads with a variety of current and voltage requirements must be supplied. This concept of architecture main advantage is reliability for final users, according to typical multibus schemes.

A hierarchical control structure is proposed to govern the system. Relative to local control strategy: average models of all PECs topologies included in the proposed DCMG are obtained by using switching functions. Passive elements values for each converter are obtained using well known procedures. Finally, feedback state controllers are design, then they are tuned from the desired polynomial based on overshoot percentage and settle time, further Ackemann method is used to obtain gains values for boost and DC-DC full bridge converters, all versions. PI classic controllers are used in Cük an DC-AC converters.

For higher supervisory control level, a EMS based in Petri net theory is proposed as coordinated control stage. EMA consist of three parts, a start-up test to define the initial state of the system. After initial state is set, connection-disconnection of source PECs is performed by set of FMSs which enable the needed transitions for the required state. In same manner, *SC* vector sets an specific FMSs to go through its states to add or shed loads PECs.

Simulation results for every converter in stand-alone operation are presented in order to validate each version performance in closed loop with feedback state controller as local control stage, here inductor ripple criterion is verified. Some trajectories of the DCMG architecture are under test to verify system operation and validate the concept. Some EMS operations are presented to validate supervisory control.

With results presented is possible to visualize that all the trajectories in DCMG are operative. In further stages of this development: Petri net EMS can be modified to reach another environmental aims or economic goals. Experimental PECs are under implementation to validate the DCMG in a phisycal way. A digital communications network can be considered in order optimize hardware CC resources. The proposed DCMG architecture is flexible and can be implemented in several kind of buildings considering peak power consumption and site loads demand.

**Author Contributions:** Methodology, J.M.-N.; Validation, R.O.-G.; Investigation, E.R.-S.; Writing—original draft, H.R.-E.; Writing—review & editing, C.G.-T. All authors have read and agreed to the published version of the manuscript.

**Funding:** This research received no external funding.

**Institutional Review Board Statement:** Not applicable.

**Informed Consent Statement:** Not applicable.

**Data Availability Statement:** Not applicable.

**Conflicts of Interest:** The authors declare no conflict of interest.

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
