# Peer review of "Novel Multibus Multivoltage Concept for DC-Microgrids in Buildings: Modeling, Design and Local Control"

_applsci, doi:10.3390/app13042405_

Round 1

Reviewer 1 Report

The article is a good application for DC Microgrid but does not present any innovation to the literature. The control methods used in the article are known and applied methods. The authors do not present an innovation for this area. Also, there are many misspellings in the article. Some of the misspellings are indicated in the attached file.  I recommend that the authors work on a control method that includes the coordination of converters with each other and increases the originality of the manuscript.

Author Response

Dear reviewer:

I´m really sorry for taking so much time for response.

Response in attached pdf file.

Reviewer 2 Report

The paper is very topical and of maximum interest to all those who work in the area of renewable energy.

The presentations are very well done, mathematically supported, and have complete graphic representations.

A scientific paper based on simulations must use accurate experimental data to validate the results. The paper must be completed with experimental data to support the conclusions.

I think the bibliography can be completed with more titles because a lot has been recently published on the paper's topic.

Put experimental data, reformulate the conclusions and complete the bibliography to obtain a very good paper.

Author Response

(The authors gave the same response as above.)

Reviewer 3 Report

The work seems to be interesting and innovating. However, some issues existed in the paper. Therefore, I suggest that the manuscript can be accepted after revision. Specific comments are as follows:

1. How about the stability and reliability of this system?

2. The advanced technology should be introduced to enrich the introduction, for example: Chinese Journal of Catalysis, 2022, 43, 2652–2664; Separation and Purification Technology, 2023, 304, 122401.

3. The writing of the manuscript can be further improved, there are some grammar mistakes.

4. Make sure all abbreviations are presented in full the first time appeared in the manuscript.

Author Response

(The authors gave the same response as above.)

Round 2

Reviewer 1 Report

1. Cük converter term should be replaced with the "Cuk converter" term. 

2. In figure 34, the values can't see correctly. The quality of the figure should be improved.

Author Response

Dear Reviewer 1.

Thanks.

Reviewer 2 Report

The paper is too long

Author Response

Dear Reviewer 2.

Thanks.
